# Heterogeneous Adversarial Play
# in Interactive Environments

**Manjie Xu** [1,2,4,5]  **Xinyi Yang** [1,2,4,5]  **Jiayu Zhan** [2,1,4,5]
mjxu25@stu.pku.edu.cn xinyiyang25@stu.pku.edu.cn jiayu.zhan@pku.edu.cn

**Wei Liang** [3,6,✉]  **Chi Zhang** [1,✉]  **Yixin Zhu** [2,1,4,5,7,✉]
liangwei@bit.edu.cn wellyzhangc@gmail.com yixin.zhu@pku.edu.cn

[1] Institute for Artificial Intelligence, Peking University
[2] School of Psychological and Cognitive Sciences, Peking University
[3] School of Computer Science & Technology, Beijing Institute of Technology
[4] State Key Lab of General AI, Peking University
[5] Beijing Key Laboratory of Behavior and Mental Health, Peking University
[6] Yangtze Delta Region Academy of Beijing Institute of Technology, Jiaxing
[7] Embodied Intelligence Lab, PKU-Wuhan Institute for Artificial Intelligence

https://sites.google.com/view/hap-learning

## Abstract

Self-play constitutes a fundamental paradigm for autonomous skill acquisition, whereby agents iteratively enhance their capabilities through self-directed environmental exploration (Silver et al., 2018). Conventional self-play frameworks exploit agent symmetry within zero-sum competitive settings (Balduzzi et al., 2019), yet this approach proves inadequate for open-ended learning scenarios characterized by inherent asymmetry. Human pedagogical systems exemplify asymmetric instructional frameworks wherein educators systematically construct challenges calibrated to individual learners' developmental trajectories (Bobbitt, 1918; Bengio et al., 2009). The principal challenge resides in operationalizing these asymmetric, adaptive pedagogical mechanisms within artificial systems capable of autonomously synthesizing appropriate curricula without predetermined task hierarchies. Here we present Heterogeneous Adversarial Play (HAP), an adversarial Automatic Curriculum Learning (ACL) framework that formalizes teacher-student interactions as a minimax optimization wherein task-generating instructor and problem-solving learner co-evolve through adversarial dynamics. In contrast to prevailing ACL methodologies that employ static curricula or unidirectional task selection mechanisms, HAP establishes a bidirectional feedback system wherein instructors continuously recalibrate task complexity in response to real-time learner performance metrics. Experimental validation across multi-task learning domains demonstrates that our framework achieves performance parity with state-of-the-art (SOTA) baselines while generating curricula that enhance learning efficacy in both artificial agents and human subjects.

## 1 Introduction

The ability to incrementally acquire and consolidate knowledge via environmental interactions—progressing from foundational concepts to sophisticated expertise—constitutes a defining characteristic of human intelligence (Elman, 1993; Rohde and Plaut, 1999; Bengio et al., 2009). Curriculum Learning (CL), as a structured pedagogical paradigm, enables humans to decompose complex tasks into manageable milestones, fostering robust comprehension and skill mastery (Bengio et al., 2009; Prideaux et al., 2003; Zhang et al., 2024c,d). Inspired by this biological precedent, machine learning researchers have endeavored to emulate progressive learning strategies for artificial agents, partic-

(a) Automatic Curriculum Learning (Teacher-Student)

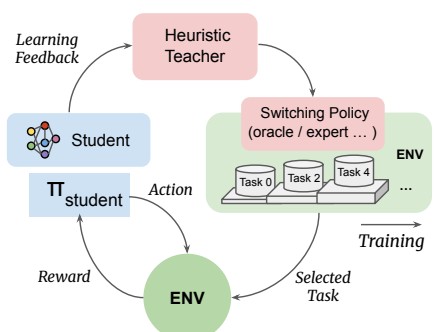

(b) Heterogeneous Adversarial Play (Ours)

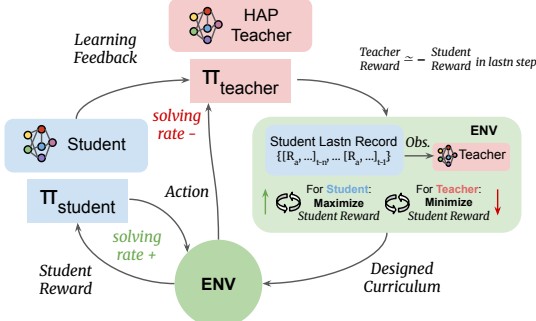

Figure 1: **Comparison of different learning frameworks.** (a) **Automatic Curriculum Learning (ACL):** A heuristic teacher selects tasks from a predefined curriculum pool and provides feedback to guide student learning through environments of increasing complexity. The teacher relies on domain expertise and rule-based policies to sequence tasks appropriately. (b) **Heterogeneous Adversarial Play (HAP):** Our framework extends ACL through adversarial co-evolution. The teacher learns to generate challenging but solvable tasks that maximize student learning, while the student adapts to solve the teacher's evolving problem proposals.

ularly in domains requiring long-horizon reasoning or multi-task proficiency (Pentina et al., 2015; Lowe et al., 2017; Narvekar and Stone, 2019; Yarats et al., 2022; Xu et al., 2023a). However, translating human pedagogical principles to Artificial Intelligence (AI) systems remains challenging due to fundamental differences in how humans and machines internalize and generalize knowledge (Khan et al., 2011; Jiang et al., 2015; Vinyals et al., 2019; Jiang et al., 2023).

Traditional CL frameworks employ static, human-designed curricula predicated on predefined task difficulty hierarchies (Graves et al., 2017; Narvekar et al., 2017; Chen et al., 2021), assuming that sequences ordered by increasing complexity universally optimize learning. This paradigm suffers from two critical limitations: (i) learners' internal states remain largely unobservable, and (ii) curricula cannot adapt dynamically to evolving capabilities. Consequently, agents may stagnate when encountering tasks that are either insufficiently challenging or prohibitively difficult, impeding exploration and convergence (Wang et al., 2021; Xu et al., 2023b). Current Automatic Curriculum Learning (ACL) methods partially ameliorate these issues by dynamically selecting tasks based on heuristic metrics (*e.g.*, success rates or loss functions) (Zhang et al., 2020; Kim et al., 2021; Portelas et al., 2021; Seo et al., 2022; Yang et al., 2024; Wu et al., 2024). However, these approaches typically operate unidirectionally, emphasizing either task generation or difficulty assessment without establishing cohesive feedback mechanisms between these components.

Cognitive science research reveals fundamental principles underlying effective curriculum design that current automated methods largely disregard. Optimal curricula necessitate individualized and adaptive task selection, embodying "hypothesis space navigation"—the systematic exploration of knowledge structures guided by learners' evolving comprehension (Tenenbaum et al., 2011). Effective instruction requires dynamic updating based on conceptual understanding models, analogous to how human educators continuously refine their mental representations of students' knowledge states (Baker et al., 2017; Chu and Schulz, 2019). Moreover, successful learning depends on bidirectional feedback loops wherein task generation and performance evaluation reciprocally inform each other (Zhu et al., 2020, 2023). These principles collectively suggest establishing productive tension between task generators that systematically challenge learners and students that continuously adapt, creating balanced adversarial dynamics characteristic of effective human developmental learning (Lake et al., 2017; Tenenbaum, 2020).

Unlike traditional self-play, which requires perfect agent symmetry (Sukhbaatar et al., 2018; Racaniere et al., 2020; Dennis et al., 2020), the teacher-student interaction naturally accommodates heterogeneous roles within an adversarial framework. We formalize this relationship as a zero-sum game wherein task generators receive rewards when problem solvers fail, while problem solvers are rewarded for successfully addressing proposed challenges. Building upon this insight, we introduce Heterogeneous Adversarial Play (HAP), an adversarial learning framework that operationalizes the challenge-response dynamics fundamental to human learning. HAP employs a dual-network architecture wherein the teacher network generates tasks calibrated to challenge student capabilities, while the student network strives to master these evolving challenges. This adversarial equilib-

rium produces curricula that dynamically balance task complexity against learners' developing proficiency, and facilitates robust knowledge consolidation as well as effective exploration.

Experimental validation across multi-task learning environments of increasing complexity demonstrates HAP's efficacy. In grid navigation domains, the framework exhibits autonomous adaptive behavior: teachers escalate task difficulty as students improve while reverting to foundational challenges when progress stagnates. In complex Minecraft-inspired environments featuring hierarchical task dependencies (Johnson et al., 2016; Hafner, 2022; Fan et al., 2022), HAP surpasses SOTA baselines in both completion rates and learning efficiency. Human studies confirm that HAP-generated curricula mirror effective pedagogical strategies, including strategic skill reinforcement and adaptive difficulty scaling. These findings indicate that adversarial co-adaptation enables robust learning in AI systems without requiring handcrafted curricula, suggesting that adversarial optimization discovers fundamental instructional principles shared across artificial and human learning systems.

Our contributions are threefold: (i) a theoretical framework grounding HAP in adversarial optimization that formalizes pedagogical interactions as minimax games, (ii) empirical validation demonstrating superior performance and efficiency compared to existing baselines across complex multi-task environments, and (iii) insights revealing alignment between machine-generated curricula and human pedagogical principles, establishing adversarial co-adaptation as a principled bridge between symmetric self-play and asymmetric curriculum learning.

## 2 Related Work

**Self-Play and Adversarial Training** Self-play has emerged as a transformative paradigm for training agents in competitive environments, whereby agents iteratively improve through interactions with evolving variants of themselves (Silver et al., 2016; Vinyals et al., 2019; Baker et al., 2019). This methodology naturally engenders curriculum formation, as progressively sophisticated agents generate increasingly challenging opponents. However, traditional self-play presupposes symmetric roles and objectives, constraining its applicability to domains requiring fundamentally heterogeneous agent capabilities (Eccles et al., 2019; Christianos et al., 2020). Recent investigations have explored asymmetric self-play configurations wherein agents assume distinct yet structurally analogous roles (Baker et al., 2019; Eccles et al., 2019; Zhang et al., 2024b). Adversarial training extends these principles by explicitly formulating interactions as zero-sum games, thereby enhancing robustness and generalization capabilities (Goodfellow et al., 2014; Ho and Ermon, 2016). Complementary approaches, including domain randomization and adversarial domain adaptation, employ adversarial objectives to facilitate transfer learning (Tzeng et al., 2017; Ganin et al., 2016; Pinto et al., 2017). While inspired by these adversarial dynamics, our HAP framework specifically addresses the inherent asymmetry between pedagogical and learning functions, transcending the symmetric constraints of conventional self-play methodologies.

**Multi-Task Learning and Transfer** Multi-task learning seeks to enhance generalization through simultaneous acquisition of multiple related tasks, exploiting shared representations and inter-task knowledge transfer (Caruana, 1997; Ruder, 2017; Crawshaw, 2020). Traditional methodologies assume static task distributions and emphasize architectural or regularization strategies to promote knowledge sharing (Misra et al., 2016; Standley et al., 2020). These approaches frequently encounter difficulties when task complexities vary substantially or when particular tasks dominate training regimes, precipitating negative transfer phenomena (Wang et al., 2019b; Fifty et al., 2021). Contemporary advances address these limitations through ACL techniques, encompassing adaptive task weighting (Chen et al., 2018; Liu et al., 2019; Yang et al., 2023), reward-based transitions (Narvekar and Stone, 2019; Chen et al., 2021; Parker-Holder et al., 2022; Li et al., 2024), and curriculum strategies governing task exposure sequences (Dennis et al., 2020; Kong et al., 2021; Soviany et al., 2022; Yang et al., 2023; Diaz et al., 2024). These ACL methodologies collectively aim to enhance sample efficiency and ultimate performance through active training guidance while enabling mastery of complex multi-goal tasks via systematic progression from elementary to advanced subtasks (Narvekar et al., 2020; Hekimoglu et al., 2023; Zhang et al., 2020; Forestier et al., 2022). Our approach builds upon this foundation (Florensa et al., 2018; Matiisen et al., 2019; Kong et al., 2021; Jiang et al., 2021) by introducing a principled mechanism for autonomous task generation and sequencing based on learner progression, simultaneously addressing task selection and difficulty calibration challenges.

**Meta-Learning and Learning to Learn** Meta-learning, or "learning to learn," endeavors to develop algorithms capable of rapid adaptation to novel tasks through exploitation of prior experience (Thrun and Pratt, 1998; Finn et al., 2017; Zhang et al., 2019). Model-agnostic meta-learning (MAML) and related variants accomplish this objective by learning initialization parameters that facilitate swift adaptation via gradient descent (Finn et al., 2017; Nichol et al., 2018; Antoniou et al., 2019). Alternative memory-based approaches utilize external memory systems or recurrent architectures to store and retrieve pertinent experiences (Santoro et al., 2016; Zhao et al., 2021; Genewein et al., 2023; Xu et al., 2025). Recent research has investigated automated curriculum generation within meta-learning contexts, wherein meta-learners identify optimal task sequences for efficient few-shot learning (Khodadadeh et al., 2019; Wang et al., 2020; Zhang et al., 2024a). Teacher-student distillation provides an additional relevant framework wherein teacher networks guide student learning through soft targets or intermediate representations (Hinton et al., 2015; Zagoruyko and Komodakis, 2017; Sengupta et al., 2023). While such methodologies focus on knowledge transfer from pre-trained instructors, HAP facilitates dynamic co-evolution wherein teachers continuously adapt to generate appropriate challenges while students concurrently develop solution capabilities, yielding a more flexible and responsive learning process.

## 3 The Heterogeneous Adversarial Play (HAP)

We formulate HAP as an adversarial optimization framework wherein a teacher agent autonomously generates challenging tasks to accelerate student learning through strategic curriculum adaptation. This asymmetric adversarial paradigm extends traditional self-play methodologies (Sukhbaatar et al., 2018) by accommodating heterogeneous agent roles with distinct capabilities and opposing objectives, thereby transcending the symmetric constraints inherent in conventional approaches.

HAP addresses fundamental limitations of manually designed curricula through continuous adaptive mechanisms. Rather than employing static task sequences that may inadequately align with student developmental trajectories, HAP implements a dynamic feedback system that modulates task difficulty in response to evolving student capabilities. Following the generative adversarial paradigm (Goodfellow et al., 2014), we model this interaction as a minimax game wherein the teacher generates progressively demanding challenges while the student systematically masters each proposed task, establishing a self-regulating curriculum that scales organically with learning progress.

### 3.1 Framework Description

We initiate our exposition with a discrete task formulation for conceptual clarity, noting that continuous extensions follow naturally. Consider a structured task space $\mathcal{T} = \{T_1, T_2, \ldots, T_N\}$ wherein each task $T_i$ exhibits varying complexity levels and potential interdependencies encoded within a directed acyclic graph $G = (\mathcal{T}, E)$. Within this environment, teacher and student agents engage in adversarial co-evolution through complementary yet opposing roles.

**Student Agent** The student agent seeks to maximize expected performance across tasks sampled from the teacher's adaptive distribution. Formally, the student optimizes policy parameters $\theta$ according to:

$$\max_{\theta} \; J_{\text{student}}(\theta) = \mathbb{E}_{T \sim p_\phi(T)} \left[ \mathbb{E}_{\tau \sim \pi(\cdot | T; \theta)} \left[ R(\tau; T) \right] \right], \tag{1}$$

where $p_\phi(T)$ represents the teacher's task selection distribution parameterized by $\phi$, $\tau$ denotes trajectories generated by the student policy $\pi(\cdot | T; \theta)$, and $R(\tau; T) = \sum_{t=0}^{H} \gamma^t r_t$ constitutes the discounted cumulative reward for task $T$.

**Teacher Agent** The teacher agent evaluates student progress and strategically modulates task selection to maintain optimal challenge levels. We implement the teacher as a neural network that processes the student's behavioral history $h_t = \{\tau_1, \tau_2, \ldots, \tau_t\}$ and outputs task selection probabilities through softmax normalization: $p_\phi(T_i | h_t) = \frac{\exp(f_\phi(T_i, h_t))}{\sum_{j=1}^{N} \exp(f_\phi(T_j, h_t))}$, where $f_\phi$ represents the teacher's scoring function. The teacher's objective directly opposes the student's performance, establishing a zero-sum adversarial dynamic:

$$\max_{\phi} \; J_{\text{teacher}}(\phi) = \mathbb{E}_{T \sim p_\phi(T)} \left[ \mathbb{E}_{\tau \sim \pi(\cdot | T; \theta)} \left[ -R(\tau; T) \right] \right]. \tag{2}$$

**Algorithm 1:** Training loop of the Heterogeneous Adversarial Play (HAP)

---
**Data:** Initial $\theta$, $\phi$; learning rates $\alpha$, $\beta$

**1** **while** *not converged* **do**

     ; /* Step 1.  Teacher's Adversarial Task Selection:           */

**2**      Generate task distribution: $p_\phi(T) \propto \exp(\phi)$;

**3**      ; /* Minimization strategy:  Sample task $T \sim p_\phi(T)$ to challenge current $\pi$ */

     ; /* Step 2.  Student's Policy Maximization:                  */

**4**      Execute $\pi(a|s, T; \theta)$, collect trajectory $\tau$;

**5**      Compute reward signal: $R(\tau; T) = \sum_{t=0}^{H} \gamma^t r_t$;

**6**      Update $\theta$ to *maximize* returns:;

**7**      $\theta \leftarrow \theta + \alpha \nabla_\theta \mathbb{E}_\tau [R(\tau; T)]$;

     ; /* Step 3.  Teacher's Adversarial Update:                     */

**8**      Update $\phi$ to *minimize* student success:;

**9**      $\phi \leftarrow \phi - \beta \nabla_\phi \mathbb{E}_T [R(\tau; T)]$;

**10**      where $\nabla_\phi J_{\text{teacher}} = -\mathbb{E}_T [\nabla_\phi \log p_\phi(T) \cdot R(\tau; T)]$;

**11** **end**

---

This adversarial formulation ensures that the teacher continuously recalibrates task difficulty in response to student capability evolution, thereby maintaining appropriate pedagogical challenge throughout the learning process.

## 3.2 Adversarial Formulation

The teacher-student interaction constitutes a minimax optimization problem that enables tractable implementation through alternating gradient-based updates:

$$\min_\phi \max_\theta \ J(\theta, \phi), \tag{3}$$

wherein the student agent maximizes expected task performance through policy optimization while the teacher agent minimizes student success by strategically selecting challenging tasks that necessitate continued adaptation.

While the student may employ any differentiable policy optimization method (*e.g.*, PPO, SAC), the teacher must adapt its parameters $\phi$ to systematically diminish student expected returns. Applying the policy gradient theorem to the teacher's task selection policy $p_\phi(T|h_t)$, we derive the gradient with respect to $\phi$ as:

$$\nabla_\phi J_{\text{teacher}}(\phi) = -\mathbb{E}_{T \sim p_\phi(T)} \left[ \nabla_\phi \log p_\phi(T) \cdot \mathbb{E}_{\tau \sim \pi(\cdot|T; \theta)} [R(\tau; T)] \right]. \tag{4}$$

This gradient formulation enables the teacher to increase the probability of selecting tasks where the student performs poorly while decreasing selection probability for tasks where the student excels. The training procedure alternates between teacher task generation, student policy execution on selected tasks, and adversarial parameter updates for both agents, as detailed in Algorithm 1.

Crucially, this adversarial framework accommodates inherently asymmetric agent roles—teachers and students possess fundamentally different capabilities, objectives, and network architectures—distinguishing HAP from traditional self-play methodologies that require agent symmetry. This asymmetric adversarial paradigm enables principled curriculum adaptation without predetermined task hierarchies, as the teacher continuously discovers optimal challenge sequences through adversarial optimization against the evolving student. Extended implementation details, including stabilization techniques and convergence analysis, are provided in Section B.

## 3.3 Implementation Details

**Cold Start Problem** Adversarial training encounters a fundamental bootstrapping challenge during initialization. The teacher possesses no prior knowledge regarding task difficulty or student capabilities, while the student begins with randomly initialized parameters yielding poor performance across all tasks. This information deficit creates an unreliable feedback mechanism that significantly impedes initial optimization progress.

We resolve this issue through a structured warm-up protocol. The student independently explores each task for a predetermined duration without teacher involvement, enabling both agents to acquire

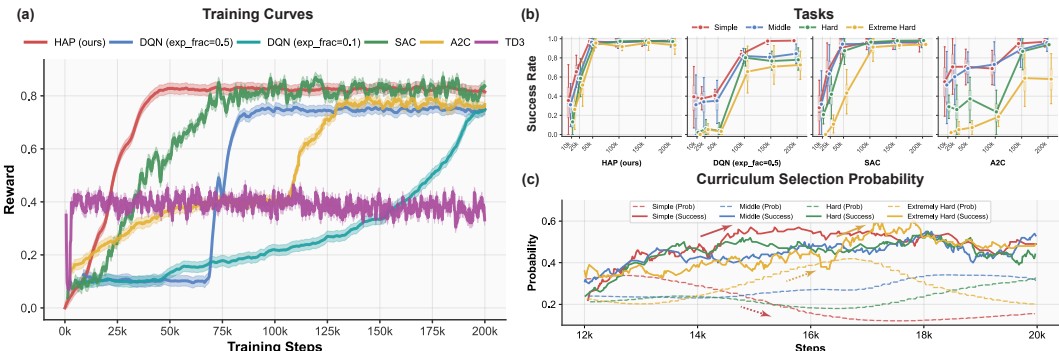

Figure 2: **Adversarial curriculum dynamics in a navigation benchmark.** (a) Training reward curves demonstrate that HAP achieves faster convergence and higher overall performance compared to baseline approaches, reaching optimal performance around 35k steps. (b) Task-specific success rates reveal that HAP learns all difficulty levels uniformly, while baselines exhibit pronounced performance gaps between easier and harder tasks throughout training. (c) Underlying adversarial mechanism: solid lines show task success rates during evaluation, while dashed lines indicate corresponding task sampling probabilities from the teacher's perspective, illustrating the positive feedback loop (increased sampling for failed tasks) and negative feedback mechanism (reduced sampling for mastered tasks) that drive HAP's effectiveness.

essential baseline information. During this phase, the student develops rudimentary competencies while the teacher observes relative task difficulties and establishes initial performance benchmarks.

**Underfitting from Task Overload**   When students attempt simultaneous learning across excessive numbers of tasks, task underfitting emerges due to insufficient attention allocation to individual challenges. This phenomenon occurs under two primary conditions: during early training when teachers maintain uniform task distributions lacking prior knowledge, and when curricula encompass numerous challenging tasks that exceed student learning capacity.

We mitigate task overload through entropy regularization of the teacher's objective:

$$J_{\text{teacher}}(\phi) = \mathbb{E}_{T \sim p_\phi(T)} \left[ \mathbb{E}_{\tau \sim \pi(\cdot|T;\theta)} \left[ -R(\tau;T) \right] \right] + \lambda \cdot \mathcal{H}(p_\phi(T)), \tag{5}$$

where $\mathcal{H}(p_\phi(T))$ denotes task distribution entropy and $\lambda$ modulates regularization intensity. This modification encourages teachers to focus student attention on manageable task subsets while maintaining appropriate exploration diversity.

**Catastrophic Forgetting**   Adversarial optimization naturally diminishes selection probability for tasks exhibiting high student success rates, redirecting focus toward more challenging objectives. While this adaptive mechanism promotes continuous learning, it may precipitate catastrophic forgetting (Kirkpatrick et al., 2017; Hadsell et al., 2020) wherein students lose proficiency on previously mastered tasks due to insufficient practice.

Although teachers could theoretically modulate selection policies within the adversarial framework to preserve task diversity, such adjustments frequently introduce training oscillations and convergence inefficiencies. We implement a more practical solution by enforcing probabilistic lower bounds on task selection, ensuring minimum exposure that prevents any task probability from approaching zero while maintaining overall curriculum balance.

### 3.4   Preliminary Experiments

We demonstrate HAP's fundamental advantages through a controlled navigation experiment designed to elucidate core adversarial dynamics. This environment requires agents to navigate from designated start positions to specified destinations using symbolic map inputs. The experimental framework encompasses four hierarchically ordered tasks—*simple*, *mid*, *hard*, and *extremely hard*—differentiated by navigation sequence length while maintaining task independence. Our teacher implementation employs learnable logits for each task, which are transformed into categorical distributions for task sampling. We evaluate HAP against established baselines using identical network architectures, conducting all experiments on a single NVIDIA A100 GPU. Detailed implementation specifications for both baselines and HAP are provided in Section C.

The reward trajectories in Figure 2(a) demonstrate that HAP achieves superior convergence efficiency, reaching optimal performance within approximately 35k training steps while sustaining the highest cumulative rewards throughout the learning process. While the environment's relative simplicity eventually enables most agents to master all tasks, TD3's convergence failure illustrates that even controlled experimental conditions pose substantial challenges for conventional Reinforcement Learning (RL) methodologies.

Task-specific success rates across training epochs, presented in Figure 2(b), reveal pronounced performance disparities between simpler and more challenging tasks under baseline approaches. This differential stems from extensive training on easier tasks leading to overfitting phenomena that subsequently impede progress on harder challenges, while task transitions precipitate catastrophic forgetting of previously acquired competencies. Conversely, HAP rapidly achieves proficiency across all tasks without exhibiting these detrimental learning pathologies.

The underlying adversarial mechanism becomes evident through Figure 2(c)'s detailed analysis of teacher-student dynamics. Solid lines denote task success rates during evaluation phases, while dashed lines indicate corresponding sampling probabilities from the teacher's selection policy. As formalized in Section 3.1, HAP operates through two synergistic feedback mechanisms: a positive reinforcement loop wherein teachers increase sampling probability for frequently failed tasks, thereby accelerating targeted skill acquisition, and a negative regulation mechanism that reduces sampling probability for mastered tasks, preventing redundant practice sessions.

These results underscore the fundamental significance of adversarial curriculum design in multi-task learning scenarios. Even within environments sufficiently tractable for eventual agent success without sophisticated intervention, dynamic task assignment regulation produces substantially smoother learning trajectories while mitigating overfitting and catastrophic forgetting. Through continuous sampling probability adjustment based on student progression, the teacher policy efficiently accelerates learning on challenging tasks while circumventing excessive repetition of mastered competencies, demonstrating that adversarial curriculum adaptation promotes both accelerated convergence and enhanced stability across diverse task distributions.

## 4 Experiments

We evaluate HAP's scalability across increasingly complex task distributions, encompassing open-world scenarios and environments featuring intricate task dependencies and interconnections.

### 4.1 Experimental Settings

**Playground** Our evaluation framework encompasses three distinct environments that span varying complexity levels and task structural characteristics. Minigrid (Chevalier-Boisvert et al., 2019) provides a highly configurable grid-world platform well-suited for examining fundamental task structures and validating core adversarial mechanisms. CRAFT (Andreas et al., 2017) represents a classic multi-task RL environment inspired by Minecraft, incorporating hierarchical crafting dependencies that necessitate systematic skill progression across interdependent subtasks. Crafter (Hafner, 2022) introduces open-world elements and stochastic dynamics, presenting substantial challenges that closely approximate real-world learning scenarios. Typical environment layouts and representative tasks are illustrated in Figure A1. Our task selection methodology prioritizes progressive difficulty scaling while ensuring meaningful dependencies between constituent subtasks. Detailed environment specifications and comprehensive task descriptions are provided in Section A.

**Baselines** We benchmark HAP against an extensive baseline suite spanning three methodological categories. Standard RL algorithms—including DQN (Mnih et al., 2013), A2C (Mnih et al., 2016), PPO (Schulman et al., 2017), SAC (Haarnoja et al., 2018), and TD3 (Fujimoto et al., 2018)—provide insights into traditional approaches' multi-task handling capabilities. DreamerV3 (Hafner et al., 2023) serves as a SOTA world model baseline, enabling assessment against contemporary model-based methodologies.

Curriculum learning approaches include Teacher–Student Curriculum Learning (TSCL) (Matiisen et al., 2019) and EXP3 auto-curriculum (Gajane et al., 2015), alongside a manually designed easy-to-hard curriculum baseline for comparative analysis. Additionally, we establish expert human

performance benchmarks through 18 trained participants possessing minimum bachelor's degree qualifications. Human performance provides an empirical upper bound reference, with participants achieving verified task mastery through rigorous pre-test qualification procedures. Complete baseline implementations and HAP specifications are detailed in Section C.

## 4.2 Quantitative Results

Table 1 establishes HAP's superior performance relative to existing algorithmic approaches across the majority of evaluated tasks and environments. In Minigrid environments, HAP achieves a general score of 0.527, surpassing all RL baselines while attaining 71% of human performance on challenging tasks. CRAFT experiments reveal particularly pronounced advantages on complex tasks, with HAP scoring 0.31 compared to DreamerV3's 0.27 on hard tasks, yielding a general score of 0.562 that narrows the human-algorithm performance gap by 30% relative to previous SOTA methodologies.

Table 1: **Performance evaluation across multi-task environments with increasing complexity.** Task success rates for algorithmic approaches compared to human experts (gray column) across Easy (basic skills), Middle (intermediate composition), and Hard (complex reasoning) difficulty levels. General scores are weighted averages across difficulties. HAP achieves superior performance on Middle and Hard tasks in Minigrid and CRAFT, with competitive results in Crafter.

| Env | Ordered | DQN | A2C | PPO | SAC | TD3 | DreamerV3 | TSCL | EXP3 | HAP | Human |
|---|---|---|---|---|---|---|---|---|---|---|---|
| **Minigrid** | | | | | | | | | | | |
| Easy | 0.67 | **0.98** | 0.94 | 0.88 | 0.97 | 0.95 | 0.96 | 0.96 | 0.97 | 0.92 | 1.00 |
| Middle | 0.12 | 0.24 | 0.25 | 0.22 | 0.27 | 0.26 | 0.34 | 0.21 | 0.24 | **0.46** | 0.78 |
| Hard | 0.20 | 0.00 | 0.00 | 0.00 | 0.13 | 0.08 | 0.18 | 0.16 | 0.18 | **0.20** | 0.46 |
| General | 0.33 | 0.407 | 0.397 | 0.367 | 0.457 | 0.43 | 0.493 | 0.443 | 0.463 | **0.527** | 0.747 |
| **CRAFT** | | | | | | | | | | | |
| Easy | 0.36 | 0.78 | 0.84 | 0.87 | 0.87 | 0.86 | 0.89 | **0.94** | 0.91 | 0.88 | 0.94 |
| Middle | 0.21 | 0.26 | 0.45 | 0.48 | 0.42 | 0.42 | 0.55 | 0.24 | 0.56 | **0.63** | 0.86 |
| Hard | 0.25 | 0.02 | 0.14 | 0.12 | 0.15 | 0.14 | 0.27 | 0.03 | 0.24 | **0.31** | 0.66 |
| General | 0.26 | 0.278 | 0.415 | 0.426 | 0.413 | 0.407 | 0.516 | 0.307 | 0.513 | **0.562** | 0.802 |
| **Crafter** | | | | | | | | | | | |
| Easy | 0.27 | 0.61 | 0.79 | **0.94** | 0.91 | 0.84 | 0.91 | 0.82 | 0.87 | 0.91 | 0.99 |
| Middle | 0.16 | 0.28 | 0.37 | 0.67 | 0.47 | 0.39 | 0.66 | 0.45 | 0.58 | **0.68** | 0.82 |
| Hard | 0.14 | 0.00 | 0.00 | 0.47 | 0.22 | 0.29 | 0.52 | 0.00 | 0.02 | **0.58** | 0.74 |
| General | 0.19 | 0.297 | 0.387 | 0.693 | 0.533 | 0.507 | 0.697 | 0.423 | 0.49 | **0.723** | 0.85 |

Crafter environments pose substantial challenges for all algorithmic approaches, wherein even HAP exhibits performance limitations despite consistently outperforming baselines on middle-difficulty tasks. These results illuminate two fundamental patterns across experimental domains. First, curriculum-based methodologies (HAP, EXP3) systematically surpass standard RL approaches on complex task configurations, demonstrating the critical importance of structured learning progression. Second, all algorithms exhibit pronounced performance degradation with increasing task complexity, contrasting sharply with human participants who maintain relatively stable performance across difficulty gradients.

## 4.3 Qualitative Analysis

Detailed examination of HAP's performance dynamics in Crafter environments (Figure 3) reveals distinctive patterns across task complexity gradients. On elementary tasks, most methodologies achieve comparable performance with consistently high success rates, indicating that conventional learning approaches possess sufficient exploration capabilities for fundamental environment interactions. However, pronounced performance disparities emerge as task complexity escalates.

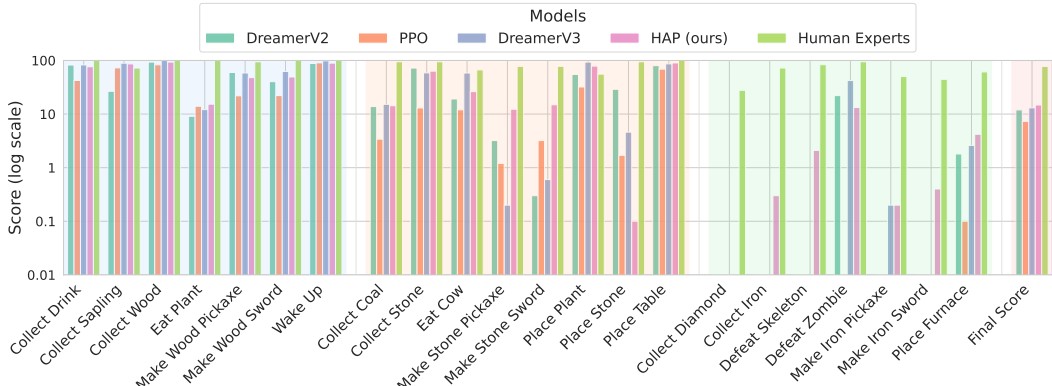

Figure 3: **Task-specific performance breakdown in Crafter.** Average scores across individual achievements calculated from task success rates. HAP performs comparably to DreamerV3 on most tasks while demonstrating superior performance on complex, multi-step challenges requiring hierarchical skill composition.

HAP demonstrates substantially elevated success rates on challenging tasks relative to baseline approaches, with advantages becoming particularly evident on complex objectives such as "Defeat Skeleton" and "Make Iron Pickaxe." These tasks necessitate sophisticated skill composition and extended planning horizons, representing quintessential challenges in hierarchical learning domains. The observed performance improvements derive from HAP's adversarial architecture, which automatically identifies and proposes prerequisite skills prior to attempting complex task execution.

This systematic prerequisite identification effectively addresses the exploration bottlenecks that fundamentally constrain standard RL methodologies in hierarchical domains. While conventional approaches struggle with the exponential search spaces inherent in complex skill composition, HAP's teacher-student dynamics naturally decomposes challenging objectives into manageable learning sequences. The adversarial framework thus enables more efficient navigation of task dependency structures, facilitating robust skill acquisition across sophisticated behavioral repertoires.

## 4.4 Discussion

Our findings illuminate critical insights regarding the current capabilities and limitations of automated curriculum learning methodologies. Despite substantial algorithmic advances, human superiority persists on extremely challenging tasks requiring sophisticated planning and compositional reasoning. Our optimal model achieves merely 65% of human performance on demanding Minigrid tasks and 47% on CRAFT environments, underscoring fundamental disparities in how humans and contemporary AI systems approach abstraction and adaptive problem-solving, particularly within compositional reasoning scenarios exemplified by CRAFT's multi-step crafting dependencies.

The effectiveness of adversarial curriculum design exhibits pronounced environment-dependent variation. HAP yields substantial improvements for challenging tasks within Minigrid and CRAFT environments, where task structures demonstrate clear hierarchical organization and well-defined dependency relationships. Conversely, performance gains prove more modest within Crafter's openworld configuration, suggesting that while adversarial curricula excel in structured task hierarchies, they may require supplementary mechanisms to support environments demanding autonomous exploration and self-directed learning strategies.

Environmental complexity introduces disproportionate performance degradation for algorithmic approaches relative to human learners. Performance declines precipitously from deterministic Minigrid environments to stochastic, partially observable Crafter scenarios, with certain baselines (TD3/PPO) achieving zero success on Crafter's challenging tasks while human participants maintain consistent overall performance levels. This algorithmic brittleness highlights fundamental limitations in handling real-world characteristics including sparse reward and delayed feedback.

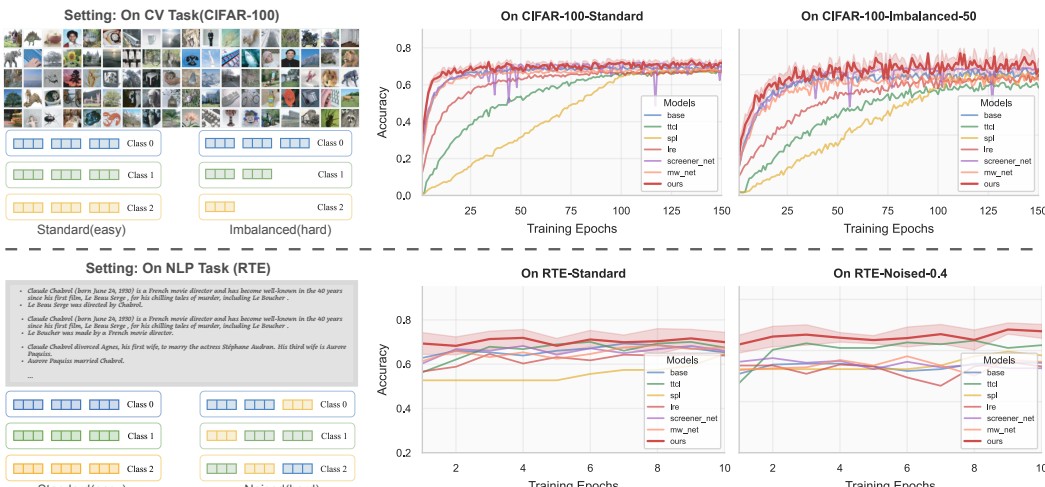

Figure 4: **Supervised curriculum learning performance on CIFAR-100 and RTE benchmarks.** HAP consistently outperforms established curriculum learning baselines across both computer vision and NLP tasks, demonstrating particular robustness in challenging scenarios with class imbalance (CIFAR-100) and label noise (RTE). Results show superior convergence speed and final performance compared to data selection and loss reweighting approaches. Min-max ranges shown for HAP for visual clarity.

# 5   Extended Evaluations

**Supervised Curriculum Learning**   Although HAP was designed for RL, we evaluate its applicability in supervised learning, where it functions as a data-level curriculum algorithm that dynamically selects training samples for performance. We assess HAP using the Curbench benchmark (Zhou et al., 2024) across computer vision (CIFAR-100 (Krizhevsky, 2009)) and natural language processing (Recognizing Textual Entailment (RTE) from GLUE (Wang et al., 2019a)). We compare against established curriculum methods in two categories: data selection approaches (TTCL (Weinshall et al., 2018), SPL (Kumar et al., 2010)) that prioritize samples by difficulty, and loss reweighting methods (LRE (Ren et al., 2018), ScreenerNet (Kim and Choi, 2018), MW-Net (Shu et al., 2019)) that adjust sample importance during training. To stress-test curriculum efficacy, we use challenging conditions: imbalanced class distribution for CIFAR-100 and noisy labels for RTE.

Figure 4 shows HAP achieves competitive performance with SOTA curriculum algorithms across both domains. Under standard conditions, HAP and most baselines approach performance upper bounds, confirming curriculum benefits are most pronounced in challenging scenarios. In demanding settings with noise or imbalance, HAP consistently outperforms most baselines, matching ScreenerNet on CIFAR-100-Imbalanced-50 and TTCL on RTE-Noised-0.4 while demonstrating faster convergence. These results indicate that adversarial curriculum principles underlying HAP generalize effectively beyond RL to supervised learning domains, supporting broader applicability of teacher-student adversarial frameworks across diverse machine learning paradigms.

**Human Study**   To validate HAP-generated curricula, we conducted a human study investigating whether adversarially optimized curricula share qualities of effective human instruction. We recruited 30 participants via Prolific and measured learning in Minigrid across three conditions: (i) no tutorial control group establishing baseline performance, (ii) expert tutorial group receiving step-by-step instruction sequences crafted by Minigrid experts, and (iii) HAP-generated tutorial group experiencing dynamically adapted curricula where the framework continuously adjusted tasks based on real-time performance. This design enables direct comparison between adversarial optimization and established pedagogical strategies.

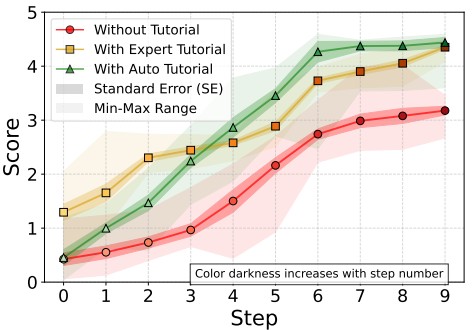

Figure 5: **Human learning performance across curriculum conditions in Minigrid.** Comparison of human subjects' learning trajectories under three conditions: no tutorial, expert step-by-step tutorial, and HAP-generated tutorial, demonstrating the effectiveness of adversarial curriculum design for human learners.

Figure 5 demonstrates that structured curricula significantly accelerate early-stage skill acquisition. Both expert-designed and HAP-generated tutorials produced similar learning rates and final performance levels, indicating that adversarial optimization successfully discovers effective pedagogical principles. While experts provided superior within-step improvement, HAP offered more individualized curricula responding to participant-specific learning patterns, resulting in faster overall performance gains across curriculum progression.

These findings suggest HAP's adversarial dynamics naturally converge toward instructional strategies consistent with effective human teaching practices, including strategic scaffolding and adaptive difficulty adjustment. The framework's ability to autonomously discover optimal instructional sequences without explicit pedagogical programming demonstrates the fundamental connection between adversarial optimization and effective curriculum design.

# 6   Conclusion

We propose Heterogeneous Adversarial Play (HAP), an adversarial learning framework where teacher and student models achieve superior task success rates and enhanced responsiveness during learning plateaus. Human studies demonstrate that HAP's curricula enhance both artificial and human learning without requiring handcrafted instruction sequences. Our work adapts dynamically to learner capabilities while maintaining pedagogical effectiveness comparable to expert instruction.

## Acknowledgments and Disclosure of Funding

This work is supported in part by the National Science and Technology Innovation 2030 Major Program (2025ZD0219402), the National Natural Science Foundation of China (32471098), the PKU-BingJi Joint Laboratory for Artificial Intelligence, and the National Comprehensive Experimental Base for Governance of Intelligent Society, Wuhan East Lake High-Tech Development Zone.

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

# A Environments

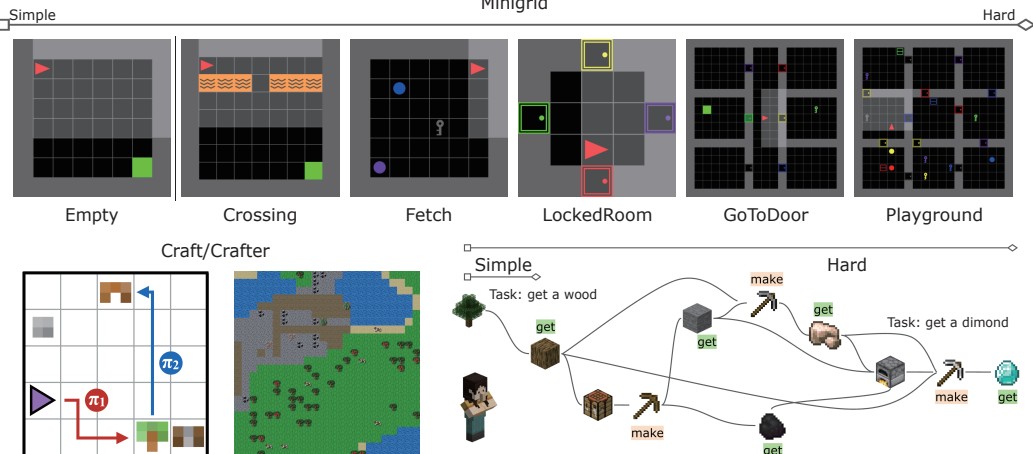

Figure A1: **Experimental environments spanning discrete navigation to open-world scenarios. Top:** Minigrid environment with six tasks arranged by increasing difficulty (left to right), from simple navigation (Empty, Crossing) to complex reasoning (Playground), enabling systematic curriculum evaluation on well-defined hierarchies. **Bottom left:** CRAFT and Crafter environments provide Minecraft-inspired multi-task scenarios with crafting mechanics and procedural generation, testing curriculum adaptation in complex domains. **Bottom right:** Task dependency graph showing hierarchical skill structure, where nodes represent individual skills and edges indicate prerequisites. Tasks with longer dependency chains present increased complexity, evaluating HAP's ability to navigate multi-step planning challenges.

We mainly include Minigrid, CRAFT, and Crafter as our testbed. We divide tasks in each benchmark into three levels: easy, middle, and hard, to test the learning progress separately. Considering factors like resource availability, crafting complexity, and or level of skill or progression required. To ensure fair comparison across methods, we customize the task structures within each environment to create unified benchmarks. Each environment contains tasks of varying difficulty levels with complex interdependencies designed to test different aspects of curriculum learning.

## A.1 Minigrid

Minigrid is a collection of 2D grid-world environments with goal-oriented tasks. Our implementation is based on the open-sourced code[1]. We include six tasks from the Minigrid task pool:

**Empty**   The agent must reach a green goal square in an empty room, with a sparse reward and a step penalty.

**Crossing**   The agent must navigate to the goal while avoiding deadly lava rivers, which have single safe crossing points.

**DoorKey**   This environment has a key that the agent must pick up in order to unlock a door and then get to the green goal square.

**FourRooms**   In this classic four-room RL environment, the agent must navigate in a maze composed of four rooms interconnected by 4 gaps in the walls. To obtain a reward, the agent must reach the green goal square. Both the agent and the goal square are randomly placed in any of the four rooms.

**MultiRoom**   This environment has a series of connected rooms with doors that must be opened in order to get to the next room. The final room has the green goal square the agent must get to.

---

[1]https://github.com/Farama-Foundation/Minigrid

**Playground**    An environment with multiple rooms and random objects. This environment originally has no specific goals or rewards. The agents are tasked to collect all objects in our research.

It has been shown that hard tasks in Minigrid can be solved in a curriculum way, *i.e.*, the MultiRoom environment can be solved by gradually increasing the number of rooms with a human-defined curriculum. However, such curricula rely on human expertise and lack generalization. We explore automated curriculum policies across diverse tasks in this work.

Table A1: **Tasks in Minigrid.**

| Level | Tasks |
|---|---|
| Easy | Empty, Crossing (Simple navigation) |
| Middle | DoorKey, FourRooms (Tool using / Multi-room) |
| Hard | MultiRoom, Playground (Larger maps with challenging tasks) |

## A.2    Craft

Table A2: **Tasks in CRAFT.**

| Level | Tasks |
|---|---|
| Easy | get[grass], get[wood], make[stick], make[plank], get[rock] |
| Middle | get[iron], get[gold], make[axe], make[bench], make[rope], make[arrow], make[knife], make[shears], make[slingshot], make[cloth] |
| Hard | get[gem], make[bed], make[bow], make[bridge], make[bundle] make[flag], make[goldarrow], make[hammer], make[ladder] |

**CRAFT**    CRAFT (CraftEnv) is a 2D crafting simulation adapted from Andreas et al. (2017), designed to support flexible, hierarchical tasks with sparse rewards in a fully procedural world. Agents must navigate, collect items, manage an inventory, and transform materials at workshops to accomplish a range of tasks. Many tasks are multi-step and require combining resources and actions in sequence—such as building a bridge to access gold—which can be challenging for agents using random exploration. The environment supports different tasks varying in complexity, from simple collection tasks to intricate multi-step crafting objectives. Our implementation is based on open source code[2].

We split the tasks in CRAFT into different levels. Easy tasks in CRAFT are straightforward, require minimal resources, and can be done early in the game with basic tools or no tools at all. Middle tasks require more resources, better tools, or intermediate crafting steps. They are achievable after some progression in the game. Tasks labeled hard are more advanced, require rare resources, or involve complex crafting chains. They are typically done later in the game. Table A2 shows all tasks in CRAFT.

Table A3: **Tasks in Crafter.**

| Level | Tasks |
|---|---|
| Easy | collect[wood], collect[sapling], eat[plant], make[wood_pickaxe] make[wood_sword], place[plant], wake_up |
| Middle | collect[stone], collect[iron], collect[coal], make[stone_pickaxe] make[stone_sword], place[stone], place[table], eat[cow] |
| Hard | collect[diamond], defeat[skeleton], collect[drink], make[iron_pickaxe] make[iron_sword], place[furnace], defeat[zombie] |

**Crafter**    The tasks in Crafter can be divided into three categories similar to those in CRAFT, as shown in Table A3. Since the original Crafter environment is too challenging for current agents, particularly in survival tasks, we removed the survival requirements to better evaluate performance on more complex tasks. This modification led to generally improved performance compared to the

---

[2]https://github.com/Feryal/craft-env

baselines reported in the original Crafter paper. For clearer comparison, the results of the baselines in Crafter have been normalized to the range $[0, 1]$. We include a fair comparison with the officially reported algorithms, plus the human-expert results reported in Achievement-Distillation, as shown in Table A4. Our implementation is based on open source code[3].

Table A4: **Performance comparison of different algorithms in Crafter.**

| Algorithm | Score (%) | Reward | Open Source |
|---|---|---|---|
| Curious Replay | 19.4$\pm$1.6 | - | AutonomousAgentsLab/cr-dv3 |
| PPO (ResNet) | 15.6$\pm$1.6 | 10.3$\pm$0.5 | snu-mllab/Achievement-Distillation |
| DreamerV3 | 14.5$\pm$1.6 | 11.7$\pm$1.9 | danijar/dreamerv3 |
| LSTM-SPCNN | 12.1$\pm$0.8 | — | astanic/crafter-ood |
| EDE | 11.7$\pm$1.0 | — | yidingjiang/ede |
| OC-SA | 11.1$\pm$0.7 | — | astanic/crafter-ood |
| DreamerV2 | 10.0$\pm$1.2 | 9.0$\pm$1.7 | danijar/dreamerv2 |
| PPO | 4.6$\pm$0.3 | 4.2$\pm$1.2 | DLR-RM/stable-baselines3 |
| Rainbow | 4.3$\pm$0.2 | 6.0$\pm$1.3 | Kaixhin/Rainbow |
| HAP (Ours) | 14.3$\pm$1.2 | 9.2$\pm$0.9 | - |
| HAP (Ours, in easier mode) | 25.1$\pm$3.2 | 15.2$\pm$2.1 | - |
| Humans (Achievement Distillation) | 50.5$\pm$6.8 | 14.3$\pm$2.3 | - |
| Humans (Ours, in easier mode) | 60.5$\pm$9.2 | 17.8$\pm$2.5 | - |

# B The Heterogeneous Adversarial Play (HAP)

Unlike traditional cooperative curriculum learning—where the teacher selects tasks in an optimal "Goldilocks zone" of difficulty to facilitate learning—we intentionally frame the teacher-student interaction in HAP as a zero-sum, adversarial process. This design centers on a dynamic equilibrium: as the student masters current tasks, the teacher autonomously generates more challenging problems, continually raising the bar and expanding the space of solvable tasks. The teacher's objective is not merely to assist learning, but to produce tasks that are maximally challenging and valuable, thereby driving the student to acquire advanced capabilities.

Formally, let $\bar{r}_{\text{stu}} = \frac{1}{n} \sum_{i=1}^{N} r_{\text{stu}}^i$ denote the student's average performance. The teacher's reward is then defined as

$$r_{\text{teacher}} = \begin{cases} 0, & \text{if } \bar{r}_{\text{stu}} = 0 \text{ or } \bar{r}_{\text{stu}} \leqslant 1 - \epsilon \\ -\bar{r}_{\text{stu}}, & \text{otherwise,} \end{cases} \tag{A1}$$

The student's reward can be structured in an analogous manner, reinforcing the zero-sum setup:

$$r_{\text{stu}} = \begin{cases} 0, & \text{if } \bar{r}_{\text{stu}} = 0 \text{ or } \bar{r}_{\text{stu}} \leqslant 1 - \epsilon \\ \bar{r}_{\text{stu}}, & \text{otherwise} \end{cases} . \tag{A2}$$

The core motivation for this adversarial framing is that, if the student consistently solves tasks, there are no further meaningful learning opportunities—the system ceases to be educationally useful. Instead, our framework ensures that teacher and student are continually co-adapting: the teacher constructs tasks just beyond the student's current ability, and the student strives to keep up. This dynamic bootstrapping not only accelerates learning progress but also encourages the emergence of both a problem proposer and a solver capable of meeting highly challenging, diverse tasks. In contrast to static, handcrafted curricula, adversarial optimization discovers and instantiates fundamental pedagogical principles underlying effective instruction in both artificial and natural systems.

This pure adversarial setup may introduce training difficulties. Ideally, the zero-sum formulation creates a dynamic equilibrium where the teacher finds tasks that maximally challenge the student's current capabilities. But for teachers like a simple probability teacher, there is indeed no such guarantee. We do encounter cases when training collapses due to pathological teacher task selection, so we further introduce entropy regularization and cold-start policies to help avoid these cases.

We provide two versions of HAP for reference:

---

[3]https://github.com/danijar/crafter

**Algorithm 2: Detailed adversarial training loop of the Heterogeneous Adversarial Play (HAP).**

---

**Data:** Initial student policy parameters $\theta_0$, teacher parameters $\phi_0$; learning rates $\alpha$, $\beta$; task set $\mathcal{T}$, rollout batch size $N$, trajectory length $H$

---

1 **for** *iteration* $k = 1, 2, \ldots, K$ **do**

    /* Observe Student History                                      */

2     Retrieve or update student behavior window $h_k$ (*e.g.*, recent returns, trajectories, or success rates);

    /* Teacher Task Distribution Computation                     */

3     Compute teacher logits $\ell = f_{\phi_{k-1}}(h_k)$;

4     Compute task probabilities $p_{\phi_{k-1}}(T_j | h_k) = \text{softmax}([\ell_j]_{j=1}^N)$;

    /* Task Sampling and Environment Setup                        */

5     Sample a mini-batch of $N$ tasks $\{T^{(i)}\}_{i=1}^N \sim p_{\phi_{k-1}}(T | h_k)$;

    /* Student Policy Rollouts                                   */

6     **for** *each task $T^{(i)}$ in the batch* **do**

7         Initialize environment in starting state $s_0 \sim \mathcal{E}(T^{(i)})$;

8         Roll out student policy $\pi_{\theta_{k-1}}(a | s, T^{(i)})$ for $H$ steps;

9         Record trajectory $\tau^{(i)} = \{(s_t, a_t, r_t)\}_{t=0}^H$;

10         Compute total (discounted) task return: $R(\tau^{(i)}; T^{(i)}) = \sum_{t=0}^H \gamma^t r_t$;

11     **end**

    /* Student Policy Update (Maximization)                     */

12     Estimate or compute advantage $\widehat{A}^{(i)}$ for each trajectory, *e.g.*, with baseline or critic;

13     $g_\theta \leftarrow \frac{1}{N} \sum_{i=1}^N \nabla_\theta \log \pi_{\theta_{k-1}}(a_{0:H}^{(i)} | s_{0:H}^{(i)}, T^{(i)}) \cdot \widehat{A}^{(i)}$;

14     Update: $\theta_k \leftarrow \theta_{k-1} + \alpha \, g_\theta$;

    /* Teacher Adversarial Update (Minimization)                */

15     $g_\phi \leftarrow -\frac{1}{N} \sum_{i=1}^N \nabla_\phi \log p_{\phi_{k-1}}(T^{(i)} | h_k) \cdot R(\tau^{(i)}; T^{(i)})$;

16     Update: $\phi_k \leftarrow \phi_{k-1} + \beta \, g_\phi$;

    /* (Optional) Logging and Evaluation                        */

17     Log statistics: average returns, task distribution, teacher entropy, etc.;

18     **if** *convergence or early stopping criteria met* **then**

19         break;

20     **end**

21 **end**

---

Or, with a simple probability teacher:

**Algorithm 3: Simple probability teacher.**

---

**Require:** Initial student policy parameters $\theta$, teacher parameters $\phi$
**Require:** Learning rates $\alpha$ (student), $\beta$ (teacher)

1 **while** *not converged* **do**
    /* Teacher Task Selection:                                */
2     Compute task probabilities using a softmax over teacher parameters:;
3     $p_\phi(T_i) = \frac{\exp(\phi_i)}{\sum_{j=1}^{N} \exp(\phi_j)}$;
4     Sample a task $T$ from the distribution $p_\phi(T)$:;
5     $T \sim p_\phi(T)$;
    /* Student Policy Execution:                          */
6     Student interacts with the environment $\mathcal{E}$ on task $T$ using policy $\pi(a \mid s, T; \theta)$;
7     Collect trajectory $\tau = \{s_0, a_0, r_0, \ldots, s_H\}$ and compute cumulative reward:;
8     $R(\tau; T) = \sum_{t=0}^{H} \gamma^t r_t$;
    /* Student Update:                                       */
9     Update student policy parameters $\theta$ to maximize expected reward:;
10     $\theta \leftarrow \theta + \alpha \nabla_\theta J_{\text{student}}(\theta)$;
11     where;
12     $J_{\text{student}}(\theta) = \mathbb{E}_{\tau \sim \pi(\cdot \mid T; \theta)} [R(\tau; T)]$;
    /* Teacher Update:                                       */
13     Compute the gradient of the teacher's objective:;
14     $\nabla_\phi J_{\text{teacher}}(\phi) = -\mathbb{E}_{T \sim p_\phi(T)} \left[ \nabla_\phi \log p_\phi(T) \cdot \mathbb{E}_{\tau \sim \pi(\cdot \mid T; \theta)} [R(\tau; T)] \right]$;
15     Update teacher parameters $\phi$ to minimize the student's expected reward:;
16     $\phi \leftarrow \phi - \beta \nabla_\phi J_{\text{teacher}}(\phi)$;
17 **end**

---

# C   Experiment Details

Table A5: **Model Parameters – Nav Task.**

| Component | Parameter | Value / Description |
|---|---|---|
| **Student Policy** | Framework | A2C |
| | Actor/Critic Hidden layers | 2 |
| | Actor/Critic Hidden units/layer | 256, 128 |
| | Activation | ReLU |
| | Optimizer | Adam |
| | Learning rate | 1e-4 |
| | Discount ($\gamma$) | 0.99 |
| | Task embedding dim | 512 |
| **Teacher Policy** | Network type | MLP |
| | Input (history vec) | Last 100 student returns |
| | Hidden layers | 2 |
| | Hidden units/layer | 256, 128 |
| | Update Freq | 1000 steps |
| | Activation | ReLU |
| | Optimizer | Adam |
| | Learning rate | 1e-4 |
| | Task Window | 4 |
| | Batch size (trajectories/update) | 32 |
| | Max steps (per episode) | 200 |

**Nav**   See Table A5.

**Minigrid**   See Table A6.

**CRAFT**   See Table A7.

**Crafter**   See Table A8.

Table A6: **Model Parameters – Minigrid Task.**

| Component | Parameter | Value / Description |
|---|---|---|
| **Student Policy** | Framework | PPO |
| | Actor/Critic Hidden layers | 2 |
| | Actor/Critic Hidden units/layer | 256, 128 |
| | Activation | ReLU |
| | Optimizer | Adam |
| | Learning rate | Policy: 3e-4; Value: 1e-3 |
| | $\gamma$ | 0.99 |
| | Task embedding dim | 512 |
| | $\epsilon$ | 0.1 |
| | GAE $\lambda$ | 0.95 |
| | ent_coef | 0.01 |
| | vf_coef | 0.5 |
| **Teacher Policy** | Network type | MLP |
| | Input (history vec) | Last 100 task indices |
| | Hidden layers | 2 |
| | Hidden units/layer | 256, 128 |
| | Update Freq | 100 episodes |
| | Activation | ReLU |
| | Optimizer | Adam |
| | Learning rate | $1 \times 10^{-3}$ |
| | Task Window | 6 |
| | Batch size (trajectories/update) | 32 |
| | Max steps (per episode) | 200 |

Table A7: Model Parameters – CRAFT Task.

| Component | Parameter | Value / Description |
|---|---|---|
| **Student Policy** | Framework | PPO |
| | Actor/Critic Hidden layers | 4 |
| | Actor/Critic Hidden units/layer | 512, 256, 256, 128 |
| | Activation | ReLU |
| | Optimizer | Adam |
| | Learning rate | Policy: 1e-4; Value: 1e-4 |
| | $\gamma$ | 0.99 |
| | Task embedding dim | 512 |
| | $\epsilon$ | 0.1 |
| | GAE $\lambda$ | 0.95 |
| | ent_coef | 0.01 |
| | vf_coef | 0.5 |
| **Teacher Policy** | Network type | MLP |
| | Input (history vec) | Last 100 task indices |
| | Hidden layers | 4 |
| | Hidden units/layer | 512, 256, 128, 128 |
| | Update Freq | 50 episodes |
| | Activation | ReLU |
| | Optimizer | Adam |
| | Learning rate | 1e-4 |
| | Task Window | 12 |
| | Batch size (trajectories/update) | 128 |
| | Max steps (per episode) | 1000 |

For the remaining baselines reported in the main draft, most are based on open-source implementations from Stable-Baselines3, the and the official CRAFT and Crafter Repo. The EXP3 baseline is re-implemented following the official blog.

Table A8: **Model Parameters – Crafter Task.**

| Component | Parameter | Value / Description |
|---|---|---|
| **Student Policy** | Framework | PPO |
| | Actor/Critic Hidden layers | 4 |
| | Actor/Critic Hidden units/layer | 512, 256, 256, 128 |
| | Activation | ReLU |
| | Optimizer | Adam |
| | Learning rate | Policy: 1e-4; Value: 1e-4 |
| | $\gamma$ | 0.99 |
| | Task embedding dim | 512 |
| | $\epsilon$ | 0.1 |
| | GAE $\lambda$ | 0.95 |
| | ent_coef | 0.01 |
| | vf_coef | 0.5 |
| **Teacher Policy** | Network type | MLP |
| | Input (history vec) | Last 100 task indices |
| | Update Freq | 50 episodes |
| | Hidden layers | 4 |
| | Hidden units/layer | 512, 256, 128, 128 |
| | Activation | ReLU |
| | Optimizer | Adam |
| | Learning rate | 1e-4 |
| | Task Window | 8 |
| | Batch size (trajectories/update) | 128 |
| | Max steps (per episode) | 1000 |

## C.1   Ablation Study: Effect of Student History on Teacher Performance

In all experiments, our teacher leverages the student's recent reward history, which we find important for assessing both current state and longer-term learning trajectories. We performed an ablation in the Minigrid environment to measure the impact of history length on overall curriculum effectiveness. Table Table A9 summarizes the results.

Table A9: **Performance in Minigrid under different history lengths for teacher. General** denotes average across all difficulty levels.

| Method | Easy | Middle | Hard | General |
|---|---|---|---|---|
| Last 1k history | 0.92 | 0.44 | 0.18 | 0.510 |
| Last 100 history | 0.92 | 0.46 | 0.20 | 0.527 |
| Without history | 0.92 | 0.43 | 0.11 | 0.487 |

Including student history leads to substantial improvements, especially in more difficult tasks and overall generalization. However, incorporating too much history (*e.g.*, last 1,000 steps), may dilute sensitivity to the student's current ability and reduce performance. Using an appropriately sized history window better reflects the learner's status and maximizes adaptive curriculum benefits.

## C.2   Teacher Update Frequency and Asynchronous Scheduling

As detailed in the tables above, the teacher's update frequency is task-dependent and tuned for each experimental setting. Typically, the teacher is updated after a fixed number of student steps, with the interval chosen to balance adaptation speed and computational efficiency. In more complex environments, updating the teacher requires a full evaluation of the student, which can introduce time bottlenecks.

We also experimented with asynchronous teacher updates, but observed no significant improvement in student performance compared to synchronous updates. Consequently, we adopt synchronous updates with carefully selected intervals to ensure efficient and effective curriculum adaptation. Further implementation details and related empirical analyses are provided in the supplementary material.

## C.3    Sensitivity Analysis and Ablation of Hyperparameters

We conducted extensive ablation studies to evaluate the impact of key hyperparameters, including entropy regularization and warm-up (cold start) duration, by running each configuration 10 times with different random seeds in the Minigrid environment. Results are reported in Table A10.

Table A10: **Performance in Minigrid under various ablation settings. General** denotes the average across all difficulty levels.

| Configuration | Easy | Middle | Hard | General |
|---|---|---|---|---|
| Original model | 0.92 | 0.46 | 0.20 | 0.527 |
| w/o entropy regularization | 0.91 | 0.38 | 0.11 | 0.467 |
| w/o cold start | 0.92 | 0.45 | 0.20 | 0.523 |
| w/o lower bounds | 3/10 | 1/10 | 0.21 | – |

Easy tasks are reliably learnable in all cases. Entropy regularization is crucial for performance on harder tasks; its removal leads to marked degradation in the 'Hard' setting. Cold start primarily impacts training efficiency, affecting convergence rate rather than final results. Lower bounds are essential for stability as models without them often fail to converge for middle and easy tasks and exhibit catastrophic forgetting on simpler tasks. This sensitivity analysis substantiates the critical role of these empirically set components in robust training across difficulty levels.

# D    Human Study

Our human study aims to verify the importance of curriculum in human learning and to evaluate the effectiveness of our algorithm in generating suitable curricula based on the human learning curve. We recruited 30 participants via the Prolific platform to ensure diverse and controlled sampling. Inclusion criteria were: age between 18 and 40, fluent English proficiency, normal or corrected-to-normal vision, and at least a bachelor's degree. Participants provided informed consent before proceeding and were compensated in accordance with institutional and Prolific guidelines. The study protocol was reviewed and monitored by a formally constituted ethics committee at our institute, in compliance with biomedical research regulations involving human subjects.

To minimize confounding factors and better align the human experimental setting with that of AI models, we modified the standard Minigrid (Chevalier-Boisvert et al., 2019) environment as follows:

**Visual Redesign**    The environment's color palette and the icons for agents, goals, and objects were replaced with high-contrast, universally interpretable symbols, but without explicit semantic meaning. This ensured that participants could not leverage any real-world prior knowledge or bias related to these elements.

**Implicit Buttons**    To intuitively guide action selection and reduce interface learning curves, all actionable elements (*e.g.*, use, toggle door, pick up) were renamed to generic labels such as "Button 1," "Button 2," *etc*. If no tutorial was provided, participants had to discover the function of each button through trial and error. However, movement buttons were made explicit, matching the clarity of available actions to the AI agents.

**Reward Design**    We adjusted the reward structure to encourage participants to maximize their score by exploring, collecting keys, opening related doors, and avoiding harmful elements. The underlying reward mechanism was not explicitly described to participants; they were only instructed to maximize their score.

**Tutorial Conditions**    We constructed three tutorial conditions for the experiment:

- **No Tutorial (Control Group):** Participants began directly in the test environment, receiving only the instruction to maximize reward.
- **Expert Step-by-Step Tutorial:** A team of Minigrid-experienced researchers manually designed an optimal skill progression—a canonical curriculum. Each mini-tutorial covered one incremental skill (*e.g.*, "navigate to goal," "unlock door," "collect target object"), and each step was presented visually and explained textually.

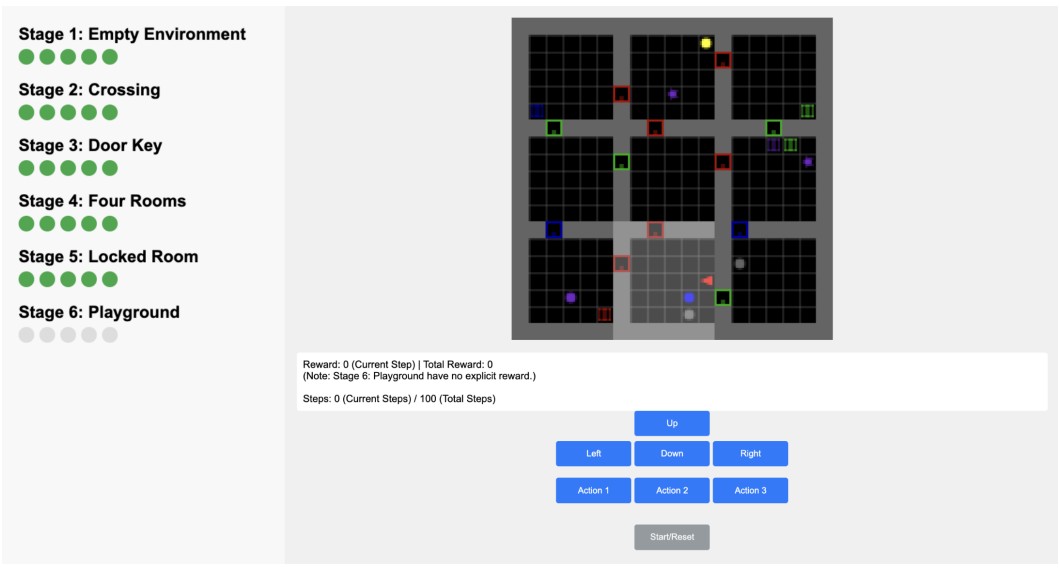

Figure A2: **Human study platform demonstration.** After completing their assigned curriculum, participants are tested in the modified Playground scenario. The left panel displays the available subtasks and illustrates the sequence of mini-tutorials provided in the Expert Step-by-Step condition.

- **AI-Generated Automatic Tutorial:** Leveraging the HAP curriculum-learning framework, we automatically generated adaptive lesson sequences. At the end of each round, the AI evaluated each participant's performance and dynamically selected the next lesson to address observed weaknesses. Note that, because the human study involved far fewer training epochs than typical AI settings, we customized HAP's hyperparameters, particularly the feedback parameters, to ensure timely and effective online adaptation of the curriculum for human learners.

Figure A2 presents a demonstration of our human study platform. After completing their assigned curriculum, all participants are ultimately tested in the modified Playground setting. The left panel shows the available subtasks and the sequence of Expert-designed Step-by-Step Tutorials.

We also set a post-experiment for the subjects, asking them about the functionality of the ambiguous buttons and elements in the game, as shown in Figure A3.

# E    Further Discussion

### E.1    Comparison with Existing Active Learning Approaches

Active Learning and Automatic Curriculum Learning (ACL) have evolved into well-developed domains with numerous sophisticated methods designed to optimize learning trajectories. While we have included a brief introduction to these related works, we would like to further elaborate on how our approach compares with existing active learning and curriculum learning methods.

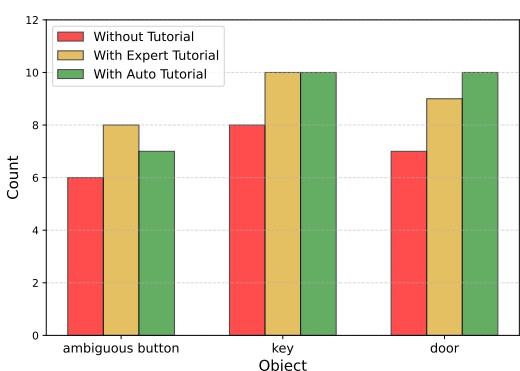

Figure A3: **Comparison of correct answers for each object type and condition in the human study.**

From the sample selection paradigm perspective, traditional active learning approaches primarily operate through sample selection based on specific criteria, measuring informativeness through uncertainty or diversity metrics. Recent advances like PORTAL (Wu et al., 2024) attempt to discover task sequences automatically but, unlike our approach, require explicit task features (specifically task similarity and difficulty metrics in PORTAL) and predefined search spaces. While these criteria effectively identify ordering relationships between tasks in controlled environments, they lack the

dynamic evaluation and generation mechanisms necessary for more complex settings. Pre-defined policies prove valuable when the task space is constrained or when learning objectives are sufficiently intuitive for human designers to create effective switching policies. However, they demonstrate significant limitations in larger, more complex environments where optimal task sequencing becomes less obvious and more context-dependent.

From the feedback loop mechanism perspective, a critical limitation in existing curriculum frameworks is their predominantly unidirectional optimization process. TeachMyAgent (Romac et al., 2023) implements mixed-difficulty curricula but relies on predetermined environment parameterizations rather than adversarially discovering optimal challenges. CurBench (Zhou et al., 2024), while establishing evaluation protocols for curriculum learning, confirms that most methods struggle with dynamic adaptation to learner progress. In contrast, our approach tries to establish a continuous bidirectional feedback loop where the teacher's task generation and the student's problem-solving capabilities co-evolve, creating a system that autonomously identifies and addresses knowledge gaps through their adversarial interaction.

Our idea draws inspiration from the remarkable success of self-play methods in artificial intelligence, particularly in complex strategic domains. AlphaGo (Silver et al., 2016) and its successors (Silver et al., 2017) demonstrated that self-play creates an emergent curriculum of increasing complexity without requiring human examples or explicit task engineering. This automatic adaptation has proven exceptionally powerful because it continually maintains an appropriate challenge level as agent capabilities evolve. More recently, similar adversarial dynamics have appeared in language model training through techniques like RLHF (Christiano et al., 2017; Ouyang et al., 2022) and adversarial prompting (Perez et al., 2022), where models improve by addressing increasingly sophisticated challenges. Our approach extends these principles beyond symmetric self-play to the inherently asymmetric teacher-student relationship, preserving the beneficial adaptive dynamics while accommodating the different roles in curriculum learning. This allows us to capture the emergent complexity benefits of adversarial approaches while tailoring the process specifically to pedagogical objectives.

## E.2  LLMs as the Teacher

Recent research has explored leveraging Large Language Models (LLMs) as curriculum designers and teachers in automated learning systems (Wang et al., 2023; Ryu et al., 2024). This emerging paradigm utilizes the extensive knowledge and reasoning capabilities of models like GPT-4 (OpenAI, 2023) and PaLM (Chowdhery et al., 2023) to generate learning tasks, provide feedback, and adapt curricula. LLM-based teachers can potentially draw on broad domain knowledge to create diverse and contextually appropriate challenges without requiring explicit programming of task generation strategies. For instance, Wang et al. (2023) demonstrated that LLMs can effectively design progressively complex tasks in Minecraft environments, while Ryu et al. (2024) showed promising results using LLMs for learning complex robot skills.

Despite these advances, we intentionally excluded LLM-based teaching approaches from our current work for several reasons. First, LLM-generated curricula, while impressive, still lack theoretical grounding in optimization principles—they operate through heuristic prompting rather than targeted adversarial dynamics. This introduces uncertainties about their ability to maintain optimal challenge levels without human oversight. Second, LLMs currently serve as task generators but typically lack integrated mechanisms to observe and adapt to learner states in real-time, creating a disconnect in the feedback loop essential to our approach. Third, the "black-box" nature of LLM-based teachers complicates analysis of emergent teaching strategies and makes it difficult to isolate the effects of curriculum design from the model's innate capabilities.

## E.3  Further extending of HAP: Meta-Learning Perspective

In the current implementation of HAP, we treat the teacher as a single neural network that takes the learner's learning performance as input and outputs the curriculum. The framework is actually simplified for easier training. We demonstrate that HAP can be extended through a meta-learning lens, where the teacher itself becomes an adaptive agent that learns optimal teaching strategies. While the original framework establishes adversarial dynamics between teacher and student, this

**Algorithm 4: Extended HAP with meta-Learning (a simple demo).**

**Data:** Initial $\theta,\phi$; learning rates $\alpha,\beta$

**1 while** *not converged* **do**

| | |
|---|---|
| | /* 1.  Observe Student's Learning State:                     */ |
| **2** | Compute teacher state $s$ from student's learning history; |
| | /* 2.  Teacher's Meta Task Generation:                        */ |
| **3** | Generate task parameters: $C = \mu_\phi(s_{\text{teacher}})$; |
| **4** | /* Using actor network to generate pedagogically valuable tasks   */ |
| | /* 3.  Student's Policy Execution:                            */ |
| **5** | Execute $\pi(a\|s, C; \theta)$, collect trajectory $\tau$; |
| **6** | Compute reward: $R(\tau; C) = \sum_{t=0}^{H} \gamma^t r_t$; |
| **7** | Update $\theta$ to maximize returns:; |
| **8** | $\theta \leftarrow \theta + \alpha\nabla_\theta\mathbb{E}_\tau[R(\tau; C)]$; |
| | /* 4.  Teacher's Meta-Learning Update:                        */ |
| **9** | Compute teacher reward $r_{\text{teacher}}$ based on student progress; |
| **10** | Store transition $(s_{\text{teacher}}, C, r_{\text{teacher}}, s'_{\text{teacher}})$ in buffer; |
| **11** | Update critic: minimize $\left(Q_\phi(s_{\text{teacher}}, C) - r_{\text{teacher}} - \gamma Q_\phi(s'_{\text{teacher}}, \mu_\phi(s'_{\text{teacher}}))\right)^2$; |
| **12** | Update actor: maximize $Q_\phi(s_{\text{teacher}}, \mu_\phi(s_{\text{teacher}}))$; |

**13 end**

extension formalizes how the teacher can systematically improve its curriculum generation through experience, albeit at a significantly higher computational cost for meta-pretraining.

In this extended formulation, we model the teacher as operating in a higher-level meta-environment where states reflect the student's learning trajectory, and actions correspond to task parameters. Unlike the simple network in our baseline approach, the teacher now employs a more sophisticated actor-critic architecture to capture the complex relationship between curriculum decisions and student progress. The teacher's state $s_{\text{teacher}}$ comprises observations about the student's learning progress, potentially including:

$$s_{\text{teacher}} = h_{\text{performance}}, h_{\text{gradients}}, h_{\text{trajectories}}, ... \tag{A3}$$

where $h$ represents historical windows of various student metrics. The teacher's meta-learning objective becomes:

$$\max_\phi J_{\text{teacher}}(\phi) = \mathbb{E}\left[\sum_t \gamma^t r_{\text{teacher},t}\right], \tag{A4}$$

where $r_{\text{teacher},t}$ incorporates pedagogical signals beyond the purely adversarial reward.

The student agent remains similar to our original formulation, learning a policy $\pi(a|s, C; \theta)$ to maximize expected returns. However, the relationship between teacher and student becomes more nuanced:

$$\max_\theta J_{\text{student}}(\theta) = \mathbb{E}C \sim \mu\phi(s_{\text{teacher}})\left[\mathbb{E}_{\tau\sim\pi(\cdot|C;\theta)}\left[R(\tau; C)\right]\right], \tag{A5}$$

where $\phi(s_{\text{teacher}})$ represents the teacher's actor network that maps the student's learning state to task parameters.

The modified training algorithm follows a similar structure to our original approach but incorporates meta-learning elements:

This meta-learning extension creates a teacher agent capable of developing sophisticated teaching strategies through experience. Unlike our baseline adversarial approach, the teacher now aims to (i) identify optimal challenge levels that maintain student engagement; (ii) recognize when to revisit foundational concepts versus introducing new challenges; (iii) develop an understanding of skill transfer and prerequisite relationships; and (iv) create coherent task sequences that build upon previously learned skills.

The original HAP framework can be seen as a rule-based version of the meta-learning extension, where we leverage adversarial policy as the teacher's intuition. The extended bidirectional learning process also mirrors sophisticated human teaching, where effective educational strategies emerge from repeated interactions rather than being fully specified in advance.

# F   Limitations and Border Impact

Our experiments were conducted on simulated learners with homogeneous skill progression patterns, which may not fully capture the complexity of real-world learning environments. Performance degradation could arise in highly heterogeneous task structures or noisy learning conditions. We cannot do real-world testing in this work due to the absence of high-fidelity simulators and the limited adaptability of baseline learning frameworks currently available. HAP can help build more intelligent AI agents, holding transformative potential for scalable, personalized adaptive AI systems, particularly in resource-constrained learning settings.

# G   Complexity Analysis of Algorithm Variants

We provide a simple analysis of the algorithmic complexity of our proposed meta-learning extension to HAP, and compare it with a simplified version to establish upper and lower bounds.

**Upper Bound: Meta-Learning Framework**   The meta-learning extension represents our upper bound in terms of computational complexity. This upper bound reflects the comprehensive nature of our meta-learning extension, which maintains detailed state representations and employs sophisticated actor-critic architectures for both teacher and student.

- **Time Complexity:** $O(|h| \cdot |s_{\text{teacher}}| + |\phi| + H \cdot |\theta| + B \cdot |\phi|^2)$ per iteration, where $|h|$ is the history length, $|s_{\text{teacher}}|$ is the dimensionality of the teacher's state representation, $|\phi|$ and $|\theta|$ are the parameter counts of teacher and student networks respectively, $H$ is the task horizon length, and $B$ is the mini-batch size for teacher updates.
- **Space Complexity:** $O(|\phi| + |\theta| + D \cdot M)$, where $D$ is the dimension of stored transitions and $M$ is the replay buffer capacity.
- **Sample Complexity:** The meta-learning approach potentially requires $O(|\mathcal{C}|^2)$ task explorations in the worst case to fully model relationships between tasks in curriculum space $\mathcal{C}$.

**Lower Bound: Simplified Algorithm**   If we replace the actor-critic architecture with basic heuristics like task cycling or simple difficulty gradients, we can get a simplified version of HAP, which maintains only minimal state about student performance (*e.g.*, success rate on the current task) and uses a predefined rule-based task selection strategy without extensive historization or predictive modeling. A simplified variant of our approach provides a lower bound on complexity:

- **Time Complexity:** $O(k + H \cdot |\theta|)$ per iteration, where $k$ is a small constant representing the complexity of a simple heuristic task selector.
- **Space Complexity:** $O(|\theta| + k')$, with $k'$ being the minimal state representation needed for basic task selection.
- **Sample Complexity:** A simple approach might require only $O(|\mathcal{C}|)$ task explorations with linear progression through the task space.

While the meta-learning approach incurs higher computational overhead, it offers significant advantages in dynamic environments with complex task interdependencies. The simplified approach may be sufficient for domains with clear, linear difficulty progression but will likely fail to identify optimal curricula in complex skill acquisition scenarios. Empirically, we observe that the additional computational cost of the meta-learning approach is justified by substantial improvements in student learning efficiency, particularly in domains where task relationships are non-obvious and student learning dynamics are complex.

