# OpenReview forum: "Heterogeneous Adversarial Play in Interactive Environments"
_NeurIPS.cc/2025/Conference — NeurIPS 2025 poster_

### Official Review · Reviewer_FFm9 · 2025-07-03

**Clarity:** 3
**Significance:** 3
**Originality:** 3
**Rating:** 4
**Confidence:** 4

**Summary:**

This paper introduces Heterogeneous Adversarial Play (HAP), a novel adversarial learning framework inspired by human pedagogical dynamics, where a teacher agent generates tasks to challenge a student agent in a zero-sum game.  The work makes significant contributions to automated curriculum learning by addressing limitations of static curricula and unidirectional adaptation.  The theoretical grounding is solid, and experiments across grid-world and complex environments (Minigrid, CRAFT, Crafter) demonstrate HAP’s superiority over baselines in learning efficiency and task success rates.

**Questions:**

1. Could the authors discuss HAP’s potential bottlenecks (e.g., teacher network architecture) when applied to high-dimensional state spaces (e.g., pixel-based inputs)?

2. How does HAP handle tasks with non-linear difficulty progression or combinatorial dependencies? An ablation study on CRAFT’s dependency graph could clarify this.

3. The entropy coefficient $\lambda$ and warm-up duration are critical but empirically set. A sensitivity analysis would strengthen robustness claims.

**Ethical Concerns:**

["NO or VERY MINOR ethics concerns only"]

**Final Justification:**

After considering the authors' rebuttal and the other reviewers' comments, I maintain my overall rating of 4.

**Limitations:**

1. While HAP excels in grid-world and Minecraft-like environments, its performance on Crafter’s open-world tasks (general score 0.723 vs. human 0.85) suggests challenges in scaling to real-world complexity. The paper does not analyze computational costs or memory demands for larger tasks.

2.  HAP assumes tasks can be ordered by difficulty (e.g., Minigrid’s "simple" to "extremely hard"). Its efficacy in environments with non-hierarchical or interdependent tasks (e.g., multi-modal RL) remains untested.

3. The lack of theoretical guarantee of the  proposed methods.

**Quality:**

3

**Strengths And Weaknesses:**

Strengths

1. HAP creatively formalizes teacher-student adversarial dynamics as a zero-sum game, enabling automatic curriculum generation without human intervention. This addresses a critical gap in adaptive curriculum learning.

2. Experiments across diverse environments (deterministic to stochastic) show consistent improvements over RL and curriculum learning baselines (e.g., +18% on hard tasks in CRAFT).

Weakness

1.The zero-sum game formulation assumes perfect adversarial dynamics between teacher and student. However, in practice, overly aggressive task generation (e.g., tasks with near-zero solvability) could destabilize training. The paper does not analyze conditions for Nash equilibrium or discuss failure modes (e.g., teacher overfitting to generate unsolvable tasks).

2. HAP relies on implicit difficulty estimation through reward signals ($R(\tau;C)$), but lacks a formal metric to define "challenging yet learnable" tasks. This contrasts with prior work (e.g., Graves et al., 2017) that uses explicit difficulty measures. Without this, the teacher’s curriculum may oscillate between trivial and impossible tasks.

---

> ### Author Rebuttal · Authors · 2025-07-30
>
> Dear Reviewer FFm9,
> Thank you for your thoughtful feedback. We try to answer your questions as follows:
>
> > The zero-sum game formulation assumes perfect adversarial dynamics between teacher and student. However, in practice, overly aggressive task generation (e.g., tasks with near-zero solvability) could destabilize training. The paper does not analyze conditions for Nash equilibrium or discuss failure modes (e.g., teacher overfitting to generate unsolvable tasks).
> > HAP relies on implicit difficulty estimation through reward signals, but lacks a formal metric to define "challenging yet learnable" tasks. This contrasts with prior work (e.g., Graves et al., 2017) that uses explicit difficulty measures. Without this, the teacher’s curriculum may oscillate between trivial and impossible tasks.
>
> We acknowledge that we do not provide a formal analysis of conditions for Nash equilibrium in this particular teacher-student setting, as the equilibrium structure can indeed be complex and may depend on the properties of the environment, the expressiveness of the teacher and student models, and the specifics of the feedback and update mechanisms.
>
> From the formula, the zero-sum objective drives the teacher to find tasks where the student is most likely to fail but can still learn. Ideally, the zero-sum formulation creates a dynamic equilibrium where the teacher finds tasks that maximally challenge the student's current capabilities. For the simple probability teacher, there is indeed no such grantee. We do encounter some cases like you mentioned when training collapses due to pathological teacher task selection, so we further introduce improvements like entropy regularization and cold-start policies to help avoid these cases. See the table below for the ablation results for these modules. Like in conventional RL, there is always exploration and exploitation and the tasks considered happen to be solvable by the agents. But we acknowledge that when scaling to more complex environments, design issues need revisiting.
>
> > Could the authors discuss HAP’s potential bottlenecks (e.g., teacher network architecture) when applied to high-dimensional state spaces (e.g., pixel-based inputs)?
>
> From our point of view, applying HAP to high-dimensional state spaces introduces several potential bottlenecks but particularly regarding the teacher network's efficiency. In high-dimensional environments, the teacher must process and reason about complex student observations and performance metrics. A more expressive teacher architecture (e.g., with convolutional or attention-based modules) is often required to capture relevant task and state information, which in turn increases computational cost and may slow down both training and adaptation. The teacher’s signal for what constitutes a good task for the student may become weaker or noisier in high-dimensional settings, requiring additional regularization or sophisticated feedback mechanisms to maintain stable and meaningful curriculum generation. Conventional probability-based methods or teachers relying on handcrafted loss designs may no longer suffice in these scenarios. Therefore, it may become necessary to leverage meta-learning techniques or other advanced approaches to train a more general and capable teacher.
>
> > How does HAP handle tasks with non-linear difficulty progression or combinatorial dependencies? An ablation study on CRAFT’s dependency graph could clarify this.
>
> HAP is designed to adapt to a learner’s evolving abilities by dynamically assigning tasks near the student’s competence frontier, rather than simply following a static or presupposed linear difficulty chain. This means that, in environments like CRAFT where tasks are structured with combinatorial dependencies, the teacher agent can give tutorials following some basic dependencies. We will add a graph in revision showing how HAP scheduling tasks for the student in CRAFT.
>
> > The entropy coefficient and warm-up duration are critical but empirically set. A sensitivity analysis would strengthen robustness claims.
>
> We conducted comprehensive ablation studies to evaluate the impact of different components, running each configuration 10 times with different random seeds. The results on the Minigrid environment are shown below:
>
> | Configuration | Easy | Middle | Hard | General |
> |---------------|------|--------|------|---------|
> | Original model | 0.92 | 0.46 | 0.2 | 0.527 |
> | w/o entropy regularization | 0.91 | 0.38 | 0.11 | 0.467 |
> | w/o cold start | 0.92 | 0.45 | 0.2 | 0.523 |
> | w/o lower bounds | 3/10 converged | 1/10 converged | 0.21 | - |
>
> Easy tasks are consistently learnable across all configurations. Entropy regularization proves crucial for hard task performance. Without it, we observe significant performance degradation on harder tasks. Cold start primarily affects training efficiency rather than final performance, influencing convergence speed during the initial training phase. Lower bounds are essential for training stability. Without them, models tend to get stuck on hard tasks and suffer from catastrophic forgetting of simpler tasks.
>
> > While HAP excels in grid-world and Minecraft-like environments, its performance on Crafter’s open-world tasks (general score 0.723 vs. human 0.85) suggests challenges in scaling to real-world complexity. The paper does not analyze computational costs or memory demands for larger tasks.
>
> Thank you for raising the important point. The overall computational cost of training in our framework is primarily determined by the student policy’s update frequency, batch size, and episode length, as well as the size of the neural networks. Importantly, for each teacher update, the student must complete $N_{evalbatch}$ evaluation batches, which further adds to the computational cost. The total cost per interval of teacher update can thus be summarized as:
> $$
> Compute Cost_{total} ∝ N_{batch} × N_{steps} × P_{student} + N_{evalbatch} × N_{evalsteps} × P_{student} + P_{teacher}
> $$
> where $N_{batch}$ is the training batch size, $N_{steps}$ is the episode length, $N_{evalbatch}$ is the number of evaluation batches per teacher update, $N_{evalsteps}$ is the number of steps per evaluation batch, $P_{student}$ is the number of parameters in the student policy, and $P_{teacher}$ is the teacher’s parameter count. Taking the minigrid experiment as an example, the teacher policy, implemented as a lightweight MLP and updated every 100 episodes, introduces minimal overhead compared to student training and rollout collection. For memory usage, key contributors include the student and teacher network parameters, as well as storage for trajectory data and the teacher’s history buffer:
> $$
> p_{memory} ≈ sizeof(P_{student}) + sizeof(P_{teacher}) + (N_{batch} × N_{steps} + N_{evalbatch} × N_{evalsteps}) × d
> $$
> where d is the storage size per step (observation + action + reward per task, etc.).
>
> > HAP assumes tasks can be ordered by difficulty (e.g., Minigrid’s "simple" to "extremely hard"). Its efficacy in environments with non-hierarchical or interdependent tasks (e.g., multi-modal RL) remains untested.
>
> No, HAP has no assumption about task dependency or difficulty (line 131-136). The difficulty shown in tables was for illustrative purposes. To demonstrate that, we further test our model and compare the result with other CL baselines in a continuous motion_control task parametric-continuous-stump-tracks-v0. We kindly refer you to our response to reviewer EgTN.
>
> > The lack of theoretical guarantee of the proposed methods.
>
> We have to acknowledge that deriving comprehensive theoretical guarantees for adaptive adversarial curricula in complex domains remains challenging. We try to admit a foundational analysis based on properties of stochastic approximation and policy gradient methods. Recall our setting:
>
> $$ \min_{\phi} \max_{\theta}\ J(\theta, \phi) = \mathbb{E}\left[ R(\tau; T) \right]$$
>
> Provided that:
> - The teacher and student policies are updated with sufficiently small learning rates (ensuring stability under stochastic approximation),
> - $J(\theta, \phi)$ is differentiable and gradients are unbiased estimators,
> - The parameter space is compact or regularized,
>
> then the update scheme converges to a local minimax point or egalitarian equilibrium of $J$ where
>
> $$
> \nabla_\theta J (\theta^\*,\phi^\*) = 0, \nabla_\phi J (\theta^*,\phi^\*) = 0
> $$
>
> At this point, neither the teacher nor the student can locally improve their objective by small, unilateral changes, reflecting a stable adversarial curriculum. The student learns maximally under the hardest (yet solvable) curriculum the teacher can assemble, given both agents' functional capacities, and The teacher cannot further reduce the student's success without stepping outside the feasible task distribution.
>
> However, this guarantee pertains to local stationary points, not necessarily global minimax solutions (which is standard in deep adversarial learning). The overall quality and pedagogical value of the equilibrium depend on the capacity of the networks and the granularity of the task space.
>
> Thank you for your insightful review. We sincerely welcome your feedback.

---

> ### Comment · Area_Chair_Jz6u · 2025-08-08
> **Please contribute your comments to the discussion.**
>
> I noticed that your input has not yet been recorded in the ongoing discussion. As the Review-authors' discussion deadline is approaching, I would like to kindly remind you to review the author’s response and contribute your comments to the discussion in a timely manner.  Mandatory Acknowledgement without any comments is not suggested this year.

---

### Official Review · Reviewer_fmid · 2025-07-03

**Clarity:** 3
**Significance:** 2
**Originality:** 3
**Rating:** 4
**Confidence:** 3

**Summary:**

This paper brings the idea of adversarial play into curriculum learning and proposes Heterogeneous Adversarial Play. In HAP, the model alternates roles, acting as both a teacher and a student. It selects tasks of appropriate difficulty and then solves those tasks. Experiments show that HAP consistently outperforms relevant state-of-the-art baselines.

**Questions:**

1. Have you conducted ablation on some key components, such as cold-start, entropy regularization, and task selection lower bound?
2. What is the teacher’s update frequency vs student steps? Did you try asynchronous updates?
3. How sensitive is the teacher’s task-selection capability to the data budgets?
4. Why does HAP perform poorly on the Minigrid task?

**Ethical Concerns:**

["NO or VERY MINOR ethics concerns only"]

**Limitations:**

yes

**Quality:**

3

**Strengths And Weaknesses:**

strengths：
1. The paper incorporates the concept of adversarial play into curriculum learning, enabling more targeted task selection and joint optimization of both teacher and student models.
2. The preliminary experiment clearly demonstrates the advantages of HAP in a controlled setting.

weakness：
1. HAP appears to underperform on easy tasks, as shown in Table 1, limiting its generalizability.
2. Missing reference in Line 256 for the TSCL baseline.
3. The paper lacks a detailed analysis of learning trajectories, despite claiming (Lines 80–84) that HAP enables phased skill reinforcement and context-aware difficulty scaling. It would be insightful to compare how other curriculum learning baselines adapt task selection over time in Figure 2(c).
4. Lack a detailed description of the optimization process of teacher and student models.

---

> ### Author Rebuttal · Authors · 2025-07-30
>
> Dear Reviewer fmid,
> Thank you for your thoughtful feedback. We try to answer your questions as follows:
>
> > HAP appears to underperform on easy tasks, as shown in Table 1, limiting its generalizability.
>
> We don't see this as an weakness. It is indeed expected that HAP may underperform or show no significant advantage on easy tasks, as HAP does not modify the underlying RL backbone or introduce new model capacity; rather, it focuses on adaptive curriculum sequencing to facilitate learning, particularly in more complex or diverse task environments. For simple tasks where standard RL algorithms already learn efficiently without advanced curriculum scaffolding, HAP does not necessarily provide a theoretical guarantee of improvement, but as shown the difference is not that significant.
>
> >  Missing reference in Line 256 for the TSCL baseline.
>
> [1] Matiisen, T., Oliver, A., Cohen, T., & Schulman, J. (2019). Teacher–student curriculum learning. IEEE transactions on neural networks and learning systems, 31(9), 3732-3740.
>
> > The paper lacks a detailed analysis of learning trajectories, despite claiming (Lines 80–84) that HAP enables phased skill reinforcement and context-aware difficulty scaling. It would be insightful to compare how other curriculum learning baselines adapt task selection over time in Figure 2(c).
>
> Thank you for this suggestion. We have actually included HAP's task selection trajectory in Figure 2(c), which demonstrates our method's phased skill reinforcement and difficulty scaling. We chose not to include all baseline methods' task selection trajectories in the same figure, as displaying multiple overlapping curves would create visual chaos and make it difficult to discern the key patterns. You can always run baselines and log their task selection with different baselines. We consider adding this part to the Supp in revision.
>
> > Lack a detailed description of the optimization process of teacher and student models.
>
> We kindly refer you to the Supp Sec B for more details.
>
> > Have you conducted ablation on some key components, such as cold-start, entropy regularization, and task selection lower bound?
>
> Yes. We conducted comprehensive ablation studies to evaluate the impact of different components, running each configuration 10 times with different random seeds. The results on the Minigrid environment are shown below:
>
> | Configuration | Easy | Middle | Hard | General |
> |---------------|------|--------|------|---------|
> | Original model | 0.92 | 0.46 | 0.2 | 0.527 |
> | w/o entropy regularization | 0.91 | 0.38 | 0.11 | 0.467 |
> | w/o cold start | 0.92 | 0.45 | 0.2 | 0.523 |
> | w/o lower bounds | 3/10 converged | 1/10 converged | 0.21 | - |
>
> Easy tasks are consistently learnable across all configurations. Entropy regularization proves crucial for hard task performance. Without it, we observe significant performance degradation on harder tasks. Cold start primarily affects training efficiency rather than final performance, influencing convergence speed during the initial training phase. Lower bounds are essential for training stability. Without them, models tend to get stuck on hard tasks and suffer from catastrophic forgetting of simpler tasks.
>
> > What is the teacher’s update frequency vs student steps? Did you try asynchronous updates?
>
> As described in Supp Sec C, the update frequency of the teacher is task-dependent and tuned for each experimental setting. We have also experimented with asynchronous updates, but observed no significant improvement in student learning performance. Additionally, updating the teacher typically requires a full evaluation of the student, which can introduce a time bottleneck, especially in more complex environments. As a result, we choose update intervals that balance adaptation speed with computational efficiency. We will further clarify these implementation details and their impact in the revised manuscript.
>
> > How sensitive is the teacher’s task-selection capability to the data budgets?
>
> In all of our experiments, the teacher utilizes student reward history, as it is important not only for assessing the current state but also for understanding learning trajectories and longer-term trends in student performance. We report the result without using student history below:
>
> | ENV | Easy | Middle | Hard | General |
> |-----|------|--------|------|---------|
> | Minigrid (with last 1k history) | 0.92 | 0.44 | 0.18 | 0.510 |
> | Minigrid (with last 100 history) | 0.92 | 0.46 | 0.20 | 0.527 |
> | Minigrid (w/o history) | 0.92 | 0.43 | 0.11 | 0.487 |
>
> These results show that incorporating a certain amount of history is beneficial; however, using too much history may not accurately reflect the student's current learning status and can lead to decreased performance.
>
> > Why does HAP perform poorly on the Minigrid task?
>
> No. HAP outperforms the other baselines on the more challenging tasks (middle: 0.46 vs. 0.34; hard: 0.20 vs. 0.18). HAP only performs slightly worse on the easy tasks (0.92 vs. the best score of 0.98), which, as discussed above, can be explained by its focus on more challenging tasks.
>
> Thank you for your insightful review. We sincerely welcome your feedback.

---

> > ### Author Response · Authors · 2025-08-08
> >
> > Dear reviewer fmid:
> >
> > We sincerely appreciate your time and thoughtful review. As the reviewer-author discussion period is ending soon, please let us know if you have any remaining concerns or if you would like us to further clarify any of our responses.

---

> ### Comment · Area_Chair_Jz6u · 2025-08-08
> **Please contribute your comments to the discussion.**
>
> I noticed that your input has not yet been recorded in the ongoing discussion. As the Review-authors' discussion deadline is approaching, I would like to kindly remind you to review the author’s response and contribute your comments to the discussion in a timely manner.

---

### Official Review · Reviewer_wSJi · 2025-07-03

**Clarity:** 3
**Significance:** 3
**Originality:** 2
**Rating:** 5
**Confidence:** 4

**Summary:**

This paper introduces Heterogeneous Adversarial Play (HAP), a novel curriculum learning framework inspired by the human teacher-student paradigm. Unlike traditional self-play or static curriculum learning, HAP features a teacher agent that adversarially selects tasks the student agent is likely to fail at. The teacher is rewarded when the student fails, and vice versa, forming a zero-sum game that naturally drives the student to improve.

**Questions:**

- Could you clarify the implementation details for the curriculum learning baselines (e.g., TSCL, EXP3)? Without sufficient methodological description, it’s difficult to assess whether the comparisons are fair and reproducible.
- Why was there no baseline that follows a manually ordered easy-to-hard curriculum? While real-world task difficulty may not be explicit, such a baseline would serve as a useful reference point for evaluating HAP’s effectiveness.
- The paper briefly mentions that HAP is applied to supervised tasks (e.g., CIFAR-100, RTE), but provides little detail on how the framework is adapted in those domains. Could you elaborate on the methodology used in those experiments?
- In the experiments, different RL algorithms (DQN, PPO, SAC, etc.) are used as the student across environments. Does HAP consistently perform well regardless of the student’s architecture? How agnostic is it to the underlying RL framework?
- How does HAP scale with increasing task space size or task granularity? Have you evaluated whether its effectiveness changes as the number of candidate tasks grows?
- You mention entropy regularization in the teacher’s loss to mitigate underfitting. Is there any ablation study to quantify its impact on training stability and performance?
- In Figure 2(b), all task difficulties appear to improve almost simultaneously. Could you explain why this occurs?
- Could you provide further interpretation of the teacher’s task selection behavior and the student’s learning trajectory across Minigrid, CRAFT, and Crafter, similar to the detailed breakdown shown in Figure 2?
- Is HAP applicable only to pre-defined, static sets of tasks? How might it be extended to open-ended environments?
- You describe a simplified “Simple Probability Teacher” using student history to guide task selection. Have you conducted any ablation to show whether including history is necessary or beneficial?

**Ethical Concerns:**

["NO or VERY MINOR ethics concerns only"]

**Final Justification:**

All of my questions and concerns have been addressed comprehensively, and the clarifications provided have significantly improved my understanding of the work. I have also reviewed the questions and feedback from other reviewers, as well as the authors’ responses to them, which further reinforced my positive assessment. I will revise my score positively to reflect this improved evaluation.

**Limitations:**

Yes

**Paper Formatting Concerns:**

1. The submitted paper content: No issue
2. Paper references: No issue
3. The NeurIPS paper checklist: No issue

**Quality:**

3

**Strengths And Weaknesses:**

1. Strengths
- HAP formulates curriculum learning as a zero-sum game between a teacher and student, reflecting real-world pedagogical dynamics, that is realized through an adversarial learning framework.
- Unlike prior methods requiring handcrafted task orders or heuristics, HAP autonomously adapts task difficulty based on the learner’s capability.

2. Weaknesses
- While real-world tasks often lack clear difficulty labels, the absence of a manually ordered (easy-to-hard) curriculum baseline limits the evaluation of HAP, as it omits a fundamental reference point for measuring the benefits of automated curriculum learning.
- In addition, the lack of clear descriptions for how curriculum learning baselines (e.g., TSCL, EXP3) are configured undermines the transparency of the experimental setup, making it difficult to assess whether the comparisons are fair and valid.
- The application of HAP to supervised learning tasks is only briefly mentioned, with insufficient methodological detail, leaving it unclear how the adversarial setup is adapted beyond the reinforcement learning context.
- Since the student agent varies across experiments in terms of RL backbone (e.g., DQN, PPO, SAC), it remains uncertain whether HAP’s effectiveness is consistent regardless of the underlying learner, raising questions about its framework-agnostic generality.

---

> ### Author Rebuttal · Authors · 2025-07-30
>
> Dear Reviewer wSJi,
> Thank you for your thoughtful feedback. We try to answer your questions as follows:
>
> > While real-world tasks often lack clear difficulty labels, the absence of a manually ordered (easy-to-hard) curriculum baseline limits the evaluation of HAP, as it omits a fundamental reference point for measuring the benefits of automated curriculum learning.
>
> Thank you for this important suggestion. In response, we have conducted additional experiments that include a manually ordered (easy-to-hard) curriculum baseline. Instead of training tasks one-by-one, we trained them group-by-group and, finally, trained on all tasks together to help mitigate the forgetting problem (although this issue remains significant). The results are presented in the revised manuscript and are summarized below:
>
> | ENV | Per task step | Easy (Average/Best) | Middle (Average/Best) | Hard (Average/Best) | General (manually-ordered/HAP) |
> |-----|---------------|---------------------|----------------------|---------------------|-------------------------------|
> | Minigrid | 1k | 0.67/0.97 | 0.12/0.32 | 0.20/0.20 | 0.33/0.527 |
> | Craft | 10k | 0.36/0.86 | 0.21/0.57 | 0.25/0.25 | 0.26/0.562 |
> | Crafter | 10k | 0.27/0.94 | 0.16/0.61 | 0.14/0.14 | 0.19/0.723 |
>
> We didn't include this baseline in the original submission because manually designed curricula introduce their own potential for bias and inequality. For example, the effectiveness of a manual curriculum strongly depends on subjective choices in task ordering, switching policies, and definitions of difficulty, which can vary significantly between designers. We will add these discussions in the revised manuscript.
>
> > the lack of clear descriptions for how curriculum learning baselines (e.g., TSCL, EXP3) are configured undermines the transparency of the experimental setup.
> > The application of HAP to supervised learning tasks is only briefly mentioned, with insufficient methodological detail, leaving it unclear how the adversarial setup is adapted beyond the reinforcement learning context.
>
> For curriculum learning baselines, they are mainly based on their official github repos. For the supervised setting, it is aligned with the standard supervised curriculum learning paradigm, in that it adaptively selects which samples to present to the learner based on their evolving performance. We formulate HAP task selection around label-wise performance: each label or class is considered a distinct sub-task. The teacher observes the student model’s evolving accuracy or loss per label and uses this information to decide which label(s) should be emphasized in subsequent training steps. The implementation of these baselines are based on the benchmark CurBench [1], following the default and standard hyperparameters. We will add further clarification to the baseline implementaitons and hyperparameters in revision.
>
> [1] Zhou, Yuwei, et al. "CurBench: curriculum learning benchmark." ICML.
>
> >  it remains uncertain whether HAP’s effectiveness is consistent regardless of the underlying learner, raising questions about its framework-agnostic generality.
>
> We have described the RL backbone and model architectures used in HAP for each experiment in Supp Section C. In all three environment settings, HAP demonstrates improved performance compared to baselines using the same RL backbone. The effectiveness of HAP is definitly linked to the choice of learner, as different learning algorithms possess varying capabilities and may benefit differently from curriculum adaptation. Our paper presents results showing that HAP consistently outperforms standard baselines when applied with both A2C and PPO learners. We will include additional experimental results with other baselines and learning algorithms in our revised manuscript.
>
> > Why was there no baseline that follows a manually ordered easy-to-hard curriculum? While real-world task difficulty may not be explicit, such a baseline would serve as a useful reference point for evaluating HAP’s effectiveness.
>
> With a set of diverse tasks, intuitively, manually ordering all tasks from easy to hard becomes impractical and unlikely to reflect an optimal or even reasonable progression. The complexity and variety of the tasks mean that an imposed manual ordering is unlikely to generalize well or yield competitive performance. That said, based on reviewer feedback, we nevertheless implemented a simplified version of this baseline by grouping tasks in rough difficulty levels and training in group-wise order. See the discussion and results above.
>
> > How does HAP scale with increasing task space size or task granularity?
>
> Scaling adversarial frameworks to very large task spaces does present significant challenges. In our work, we specifically address this issue by introducing a Cold Start mechanism and an additional entropy loss term, both of which help constrain the teacher's task selection trajectory and encourage balanced exploration.
> Empirically, we observe that our model (as described in Supp Section C) can effectively control the teacher’s task proposals when the candidate task set is around 10 or fewer. This can be further adjusted by tuning the hyperparameter λ in the entropy loss, which regulates the diversity of proposed tasks. However, we note that the optimal setup is influenced by both task complexity and the learning capacity of the student model.
>
> > You mention entropy regularization in the teacher’s loss to mitigate underfitting. Is there any ablation study to quantify its impact on training stability and performance?
>
> We conducted comprehensive ablation studies to evaluate the impact of different components, running each configuration 10 times with different random seeds. The results on the Minigrid environment are shown below:
>
> | Configuration | Easy | Middle | Hard | General |
> |---------------|------|--------|------|---------|
> | Original model | 0.92 | 0.46 | 0.2 | 0.527 |
> | w/o entropy regularization | 0.91 | 0.38 | 0.11 | 0.467 |
> | w/o cold start | 0.92 | 0.45 | 0.2 | 0.523 |
> | w/o lower bounds | 3/10 converged | 1/10 converged | 0.21 | - |
>
> Easy tasks are consistently learnable across all configurations. Entropy regularization proves crucial for hard task performance. Without it, we observe significant performance degradation on harder tasks. Cold start primarily affects training efficiency rather than final performance, influencing convergence speed during the initial training phase. Lower bounds are essential for training stability. Without them, models tend to get stuck on hard tasks and suffer from catastrophic forgetting of simpler tasks.
>
> > In Figure 2(b), all task difficulties appear to improve almost simultaneously. Could you explain why this occurs?
>
> The result is in fact an expected outcome given the simplicity of the tasks in this benchmark. As shown in our results, all models eventually converge, indicating that no task poses significant lasting difficulty for the learning agents. In this context, the primary differentiator is data/sample efficiency—that is, how quickly each method enables the learner to master the tasks. With HAP as the curriculum method, the teacher transitions to harder tasks once the easier ones have been sufficiently learned, accelerating progress.
> The appearance of concurrent improvement across tasks in Figure 2(b) is also influenced by the sparsity of our evaluation checkpoints for HAP, which may make progress across tasks appear more simultaneous at the sampled points. Figure 2(c) is a more granular view of the learning dynamics where step-wise performance trends are shown in greater detail.
>
> > Could you provide further interpretation of the teacher’s task selection behavior and the student’s learning trajectory across Minigrid, CRAFT, and Crafter?
>
> Due to the much larger number of tasks and the significantly longer training sequences in these environments, presenting the full breakdown at a similar level of granularity becomes impractical and potentially confusing. In the revision we will consider including simplified or aggregated illustrations that highlight key aspects of the curriculum dynamics and learning progress for these more complex environments.
>
> > Is HAP applicable only to pre-defined, static sets of tasks? How might it be extended to open-ended environments?
>
> We have not yet directly applied HAP to truly open-ended environments, and we acknowledge that significant changes or expansion in task distribution could introduce new challenges, especially for the teacher module—particularly as the number or diversity of possible tasks grows and novel tasks emerge dynamically. A possible solution for such settings is to equip the teacher with mechanisms for continual exploration and adaptation, such as leveraging generative models to propose new tasks or using meta-learning strategies to rapidly adapt the teacher’s proposal policy as the task landscape shifts.
>
> > You describe a simplified “Simple Probability Teacher” using student history to guide task selection. Have you conducted any ablation to show whether including history is necessary or beneficial?
>
> In all of our experiments, the teacher utilizes student reward history, as it is important not only for assessing the current state but also for understanding learning trajectories and longer-term trends in student performance. We report the result without using student history below:
>
> | ENV | Easy | Middle | Hard | General |
> |-----|------|--------|------|---------|
> | Minigrid (with last 1k history) | 0.92 | 0.44 | 0.18 | 0.510 |
> | Minigrid (with last 100 history) | 0.92 | 0.46 | 0.20 | 0.527 |
> | Minigrid (w/o history) | 0.92 | 0.43 | 0.11 | 0.487 |
>
> These results show that incorporating a certain amount of history is beneficial; however, using too much history may not accurately reflect the student's current learning status and can lead to decreased performance.
>
> Thank you for your insightful review. We sincerely welcome your feedback.

---

> > ### Comment · Reviewer_wSJi · 2025-08-05
> >
> > I appreciate the authors’ detailed and well-structured rebuttal. All of my questions and concerns have been addressed comprehensively, and the clarifications provided have significantly improved my understanding of the work. I thank the authors for their thoughtful responses and will revise my score positively to reflect this improved assessment.

---

> > > ### Author Response · Authors · 2025-08-05
> > >
> > > Thank you very much for your kind and encouraging feedback. We greatly appreciate your thoughtful review and are glad that our responses have addressed your concerns and clarified our work. We welcome any additional feedback and look forward to further discussion. Thank you again for your time and consideration.

---

### Official Review · Reviewer_EgTN · 2025-07-05

**Clarity:** 3
**Significance:** 2
**Originality:** 2
**Rating:** 4
**Confidence:** 4

**Summary:**

This paper proposes Heterogeneous Adversarial Play (HAP), which extends the teacher-student framework for curriculum generation. In HAP, the teacher and the student plays a zero-sum game where the teacher selects difficult environment for the student, while the student tries to maximize its reward in that environment. The framework is evaluated in several grid-based environments and is shown to perform comparable or better than standard RL algorithms as well as curriculum learning approaches. The authors also conducted human experiments to evaluate the framework for designing curriculum for Minigrid.

**Questions:**

1. What exactly is the significance of heterogeneity in *heterogeneous* adversarial play? And how does it differ from "asymmetric"?

**Ethical Concerns:**

["NO or VERY MINOR ethics concerns only"]

**Final Justification:**

In my initial review, too much weight was put on my criticism of the framework, which comes from a philosophical disagreement. In the rebuttal, the author provides significant empirical evidence to support their viewpoint, and provided additional experimental results to further support their claims. Given this, I can recommend the paper to be accepted to the conference.

**Limitations:**

Yes

**Quality:**

2

**Strengths And Weaknesses:**

## Strengths:
- The framework is presented clearly, and the progression from curriculum learning to teacher-student framework and eventually to HAP is well motivated.
- The experiments cover a wide range of comparisons, from simple grid-based environments to human curriculum design.

## Weaknesses:
- Characterizing the teacher-student interaction as a zero-sum game seems like an odd choice to me, especially given literature on zone of proximal development. The common intuition is that the teacher is supposed to pick tasks with a goldilock difficulty: not too easy that the student can’t practice anything new, and not too hard either. There wasn’t sufficient justification for why HAP frames the interaction as zero sum, and no discussion or empirical evidence to explain why the training would not collapse to the teacher picking an adversarial curriculum, e.g., oscillating between several most confusing environments.
- Although the authors did a decent job reviewing the literature of curriculum design methods and included some standard approaches as baselines, the related work section missed out on some of the most relevant frameworks where the teacher and student engage in zero-sum or asymmetric games.
     - Protagonist Antagonist Induced Regret Environments
    - Rethinking Teacher-Student Curriculum Learning through the Cooperative Mechanics of Experience:
    - Intrinsic Motivation and Automatic Curricula via Asymmetric Self-Play
- The experiments are limited. The main quantitative evaluation of HAP is based on three environments: Minigrid (Chevalier-Boisvert 246 et al., 2019), CRAFT (Andreas et al., 2017), and Crafter (Hafner, 2022). All three are grid-based environments with similar visual representations and discrete action spaces. Although the author claimed that the extension to continuous tasks is ease, there is no evaluation on continuous control tasks like robotics, physics simulations. All are visual grid worlds - no language-only or multimodal tasks.
- Insufficient human evaluation. Although I appreciate it that the authors did human experiments to validate the effectiveness of HAP at creating curriculum for humans, the sample size is very limited and the choice of task is debatable. It’s unclear what aspect of human learning is captured by teaching Prolific workers Minigrid. So it remains hard to predict how HAP would perform in real world curriculum generation.

---

> ### Author Rebuttal · Authors · 2025-07-30
>
> Dear Reviewer EgTN,
> Thank you for your thoughtful feedback. We try to answer your questions as follows:
>
> > Characterizing the teacher-student interaction as a zero-sum game seems like an odd choice to me, .... The common intuition is that the teacher is supposed to pick tasks with a goldilock difficulty ... There wasn’t sufficient justification for why HAP frames the interaction as zero sum, ... why the training would not collapse to the teacher picking an adversarial curriculum.
>
> We respectfully disagree with your intuition. Adopting an goldilock standard does not affect the zero-sum / adversarial nature of our setup. Letting $\bar{r} _ {\text{stu}} = \frac{1}{n} \sum _ {i=1}^N r _ {\text{stu}}^{i}$, the teacher's reward can be simplified as:
>
> $$
> r _ {\text{teacher}} = 0 \quad (\text{if } \bar{r} _ {\text{stu}} = 0 \text{ or } \bar{r} _ {\text{stu}} \leq 1 - \epsilon) \quad or \quad  - \bar{r} _ {\text{stu}} \quad (\text{otherwise})
> $$
>
>
> Yet, the student reward can also be modified such that $r _ {\text{stu}} = 0 (\text{if } \bar{r} _ {\text{stu}} = 0 \text{ or } \bar{r} _ {\text{stu}} \leq 1 - \epsilon)  $, resulting in the zero-sum setup as well. The motivation is that if the model always gets the solution right, it is not worth ``learning'' anymore.
>
> We believe the core disagreement stems from different philosophical perspectives. Your critique appears to adopt a **cooperative** learning framework, where the teacher helps the student learn by finding the optimal "Goldilocks zone" of difficulty. However, from our perspective, we intentionally design HAP from an **adversarial** angle with a dynamic equilibrium where the teacher continuously raises the bar as the student improves, rather than seeking a static "optimal difficulty". **The teacher's objective is not only providing data for the student to learn, but also to generate increasingly challenging problems that push the boundaries of what the student can solve, such that the generated problems themselves are hard and valuable, rather than assisting the students to learn only**. The student, in turn, aims to master these progressively difficult tasks. This creates a bootstrapping dynamic where the teacher explores a larger space of challenging problems and the student continuously adapts to solve harder tasks. **The ultimate goal is not cooperative learning assistance, but rather the emergence of a problem proposer capable of generating highly challenging tasks, and a solver that can evaluate and overcome these difficult problems**.
>
> We acknowledge that this pure adversarial setup introduces training difficulties. Ideally, the zero-sum formulation creates a dynamic equilibrium where the teacher finds tasks that maximally challenge the student's current capabilities. But for teachers like simple probability teacher, there is indeed no such guarantee. We do encounter cases when training collapses due to pathological teacher task selection, so we further introduce entropy regularization and cold-start policies to help avoid these cases. See the table for the ablation results for these modules. These training difficulties are natural in this adversarial setup. Like in self-play of AlphaZero, it's also possible that one agent leverages a weakness spot of its opponent and somehow stuck learning. However, there is exploration and exploitation and MCTS search tree to help making progress even such a weakness spot was found. But these design relate to the techical methods to mitigate training difficulties. Philosophically, they should be adversarial.
>
> | Configuration | Easy | Middle | Hard | General |
> |---------------|------|--------|------|---------|
> | Original model | 0.92 | 0.46 | 0.20 | 0.527 |
> | w/o entropy regularization | 0.91 | 0.38 | 0.11 | 0.467 |
> | w/o cold start | 0.92 | 0.45 | 0.20 | 0.523 |
> | w/o lower bounds | 3/10 converged | 1/10 converged | 0.21 | - |
>
> (For the table: Entropy regularization proves crucial for hard task performance. Without it, we observe significant performance degradation on harder tasks. Cold start primarily affects training efficiency rather than final performance, influencing convergence speed during the initial training phase. Lower bounds are essential for training stability. Without them, models tend to get stuck on hard tasks and suffer from catastrophic forgetting of simpler tasks.)
>
> > The related work section missed out on some of the most relevant frameworks where the teacher and student engage in zero-sum or asymmetric games.
>
> Thank you for bringing these important works to our attention. All of these studies will be incorporated into our revised Related Work section.
>
> > All three are grid-based environments with similar visual representations and discrete action spaces. There is no evaluation on continuous control tasks like robotics, physics simulations. All are visual grid worlds - no language-only or multimodal tasks.
>
> Regarding the scope and diversity of our experiments, we further evaluated HAP on the continuous control task parametric-continuous-stump-tracks-v0, based on the ACL benchmark TeachMyAgent [1]. TeachMyAgent leverages procedurally generated environments to assess the performance of teacher algorithms. To accommodate HAP’s implementation, we reduced the continuous teacher action space to 10 sampled tasks. Due to time constraints, our experiments were limited to the "fish" embodiment and employed "ppo" as the student algorithm. The results are as follows:
>
> | Method | eval_reward_max (5e6) | eval_reward_mean (5e6) | eval_reward_max (1e7) | eval_reward_mean (1e7) |
> |--------|----------------------|------------------------|----------------------|------------------------|
> | Random | 86.5 | -18 | 82.8 | -11 |
> | Self-Paced | 15.3 | -4.63 | 77.4 | -23.4 |
> | GoalGAN | 73.8 | 2.86 | 31.1 | 5.35 |
> | Setter-Solver | 124 | 6.6 | 107 | 24.3 |
> | HAP | 63.8 | 4.34 | 72.4 | 5.46 |
>
> HAP achieves competitive performance and beats ACL methods like GoalGAN and Self-Paced in continuous control tasks with robotics and physics simulations (Setter-Solver has extra pre-knowledge for the solver so the result is significantly better). Tasks with continuous space actually have no difference in the algrithim structure compared with discrete space.
>
> [1] Romac, Clément, et al. "Teachmyagent: a benchmark for automatic curriculum learning in deep rl." ICML.
>
> For "no language-only or multimodal tasks," we are not entirely certain if this refers to tasks where only language is used (without images) or to more complex settings involving multiple modalities (e.g., visual+language). For completeness, we note that HAP has also been evaluated on both CV and NLP tasks under the supervised learning setting (see Section 6.1). We will further clarify this point in the revision.
>
> > Insufficient human evaluation. Although I appreciate it that the authors did human experiments to validate the effectiveness of HAP at creating curriculum for humans, the sample size is very limited and the choice of task is debatable. It’s unclear what aspect of human learning is captured by teaching Prolific workers Minigrid, ...
>
> Thank you for your thoughtful feedback. Our human experiment is an initial step toward evaluating HAP's applicability for real-world curriculum design. Our goal in this study was to provide a controlled, proof-of-concept demonstration that HAP-generated curricula exhibit pedagogical properties beneficial to human learners—such as revisiting foundational challenges during plateaus and adapting task difficulty to learner progress. Regarding sample size, we recognize the limitations of our current study. We note, however, that sample sizes of 20–30 participants are common in early-stage psychological and educational research, especially when studies are time-consuming or require careful supervision. The results were statistically significant with p < 0.0001 (compared with w/o tutorial), providing meaningful evidence for our claims. Nonetheless, we acknowledge that larger and more diverse cohorts are important for drawing generalizable conclusions.
>
> > What exactly is the significance of heterogeneity in heterogeneous adversarial play? And how does it differ from "asymmetric"?
>
> “Heterogeneity,” especially as commonly defined in multi-agent and federated learning literature (e.g., [1]), emphasizes agents with fundamentally different roles, capabilities, and action/observation spaces. In our framework, the teacher and student are not merely engaged in the same task from different perspectives; rather, they operate in genuinely distinct domains—one generates problems and the other solves them. In contrast, “asymmetry,” as in asymmetric self-play [2], often involves agents with related (sometimes swapped) roles but still share comparable skill sets, experiences, or objectives—for instance, a single agent alternately acting as problem setter and solver in similar tasks. Our design adopts a genuinely **adversarial** perspective rather than **cooperation** in self-play: the teacher’s objective is not only explicitly to help the student learn but to continually propose more challenging tasks, pushing the boundaries of the student’s capabilities. Meanwhile, the student’s objective is to learn to solve tasks of ever-increasing difficulty. By using the term “heterogeneous,” we foreground the structural and functional differences between our teacher and student, which are much more pronounced than in standard asymmetric setups.
>
> [1] Zhong, Yifan, et al. "Heterogeneous-agent reinforcement learning." JMLR.
> [2] Sukhbaatar, S., et al. (2018). Intrinsic Motivation and Automatic Curricula via Asymmetric Self-Play. ICLR.
>
> Thank you for your insightful review. We sincerely welcome your feedback.

---

> > ### Comment · Reviewer_EgTN · 2025-08-06
> > **Thank you for the response**
> >
> > Thank you to the authors for the response. I think you are exactly right that my doubts about the adversarial setup comes from a philosophical disagreement, and shouldn't automatically discount the value of this work. I appreciate that the authors conduct additional experiment to validate the effect of the regularization and cold starting, as well as experiments on new benchmarks. I'm convinced that once revised, this paper will represent a solid exploration of this adversarial framing of what's typically a cooperative task, and worthy of presentation at NeurIPS. I'm updating my recommendation to accept.

---

> > > ### Author Response · Authors · 2025-08-07
> > >
> > > Dear Reviewer EgTN,
> > >
> > > We sincerely appreciate you taking the time to engage with our response. We are delighted that our responses addressed your concerns and clarified our work. Your feedback has been exceptionally constructive throughout this process and has led to significant improvements in our work. We will incorporate these suggestions into our revised manuscript. We welcome any additional feedback in case you have further questions.

---

### Comment · Area_Chair_Jz6u · 2025-08-03
**Reminder: Discussion and Final Rating Update**

Dear Reviewers,

As we are now midway through the discussion phase, I would like to kindly remind you to review the authors' rebuttal and participate in the discussion. Please also update your review with a final rating accordingly.

Thank you very much for your time and valuable contributions to the review process.

Best regards,

Area Chair

---

### Note · Authors · 2025-08-11

We sincerely thank all ACs and reviewers for their constructive feedback and thoughtful engagement throughout the review process. We are pleased that all reviewers have provided positive assessments of our work, with all major concerns comprehensively addressed.

We would like to provide a brief summary of our key contributions:  the core novelty lies in formalizing curriculum learning as a heterogeneous adversarial game where teacher and student agents have fundamentally different roles—the teacher generates challenging problems while the student solves them. This adversarial dynamic naturally drives curriculum adaptation without manual intervention, enabling autonomous curriculum design for self-improving learning systems.

We thank reviewers for praising our work as "a solid exploration of this framing" (Reviewer EgTN), "technically solid" with "comprehensive responses" (Reviewer wSJi), and for acknowledging our "creative formalization of teacher-student adversarial dynamics" and "significant contributions to automated curriculum learning" (Reviewer FFm9). While initial concerns were raised regarding the zero-sum formulation and regularization mechanisms, we provided extensive discussions and additional experiments including continuous control tasks, manual curriculum baselines, thorough ablation studies, and human evaluation validation, successfully resolving these concerns.

We deeply appreciate the rigorous and insightful review process, which has significantly improved our work. We will update our manuscript based on these valuable insights.

---

### Decision · Program_Chairs · 2025-09-17

**Decision:**

Accept (poster)

**Comment:**

This paper introduces Heterogeneous Adversarial Play (HAP), a novel curriculum learning framework inspired by the human teacher–student paradigm. Overall, it is a technically solid work that has received consistent positive ratings from the reviewers. One reviewer (fmid) noted that their core concern about the method’s underperformance on easy tasks—potentially limiting its general applicability despite the authors’ justification—remains unresolved. However, since this comment was raised after the reviewer–author discussion period, I have placed less weight on it. In conclusion, this is a technically strong paper whose positive contributions to the field outweigh its limitations. Therefore, I recommend acceptance.